# Importance of glutamine in synaptic vesicles revealed by functional studies of *SLC6A17* and its mutations pathogenic for intellectual disability

Xiaobo Jia[1,2,3†], Jiemin Zhu[4†], Xiling Bian[4†], Sulin Liu[2], Sihan Yu[1], Wenjun Liang[2], Lifen Jiang[5], Renbo Mao[1], Wenxia Zhang[4], Yi Rao[1,2,3,4,5,6]*‡

[1]Chinese Institute for Brain Research, Beijing, China; [2]Changping Laboratory, Beijing, China; [3]Research Unit of Medical Neurobiology, Chinese Academy of Medical Sciences, Beijing, China; [4]Laboratory of Neurochemical Biology, PKU-IDG/McGovern Institute for Brain Research, Peking-Tsinghua Center for Life Sciences, School of Life Sciences, Department of Molecular and Cellular Pharmacology, School of Pharmaceutical Sciences, School of Chemistry and Chemical Engineering, Peking University, Beijing, China; [5]Institute of Molecular Physiology, Shenzhen Bay Laboratory, Shenzhen, China; [6]Capital Medical University, Beijing, China

*For correspondence:
yrao@pku.edu.cn

†These authors contributed equally to this work

‡Lead contact

**Abstract** Human mutations in the gene encoding the solute carrier (SLC) 6A17 caused intellectual disability (ID). The physiological role of *SLC6A17* and pathogenesis of *SLC6A17*-based-ID were both unclear. Here, we report learning deficits in *Slc6a17* knockout and point mutant mice. Biochemistry, proteomic, and electron microscopy (EM) support SLC6A17 protein localization in synaptic vesicles (SVs). Chemical analysis of SVs by liquid chromatography coupled to mass spectrometry (LC-MS) revealed glutamine (Gln) in SVs containing SLC6A17. Virally mediated overexpression of SLC6A17 increased Gln in SVs. Either genetic or virally mediated targeting of *Slc6a17* reduced Gln in SVs. One ID mutation caused SLC6A17 mislocalization while the other caused defective Gln transport. Multidisciplinary approaches with seven types of genetically modified mice have shown Gln as an endogenous substrate of SLC6A17, uncovered Gln as a new molecule in SVs, established the necessary and sufficient roles of SLC6A17 in Gln transport into SVs, and suggested SV Gln decrease as the key pathogenetic mechanism in human ID.

## eLife assessment

This study makes a **valuable** contribution to our functional understanding of the atypical amino acid transporter SLC6A177 at nerve cell synapses and the role of SLC6A17 variants in certain forms of intellectual disability. The reported evidence that disease-linked SLC6A17 variants cause behavioral abnormalities is **convincing**. However, corresponding molecular underpinnings, that is, the molecular role of SLC6A17 in synapses and the functional molecular consequences of disease-related SLC6A17 variations, remain unclear because corresponding informative experimental approaches are missing – most importantly direct measurements of the transport activity of SLC6A17 in the various genetic contexts studied. This limits the robustness and validity of key mechanistic conclusions drawn from the present work.

## Introduction

With an approximate prevalence of 1%, intellectual disability (ID) in humans is a neurodevelopmental disorder with debilitating effects on patients (*Maulik et al., 2011*). Molecular genetic research into ID has identified multiple genes whose defects are underlying ID (*de Ligt et al., 2012*; *Gilissen et al., 2014*; *Hamdan et al., 2014*; *Hu et al., 2019*; *Khan et al., 2016*; *Lelieveld et al., 2016*; *Rauch et al., 2012*; *Ropers, 2010*; *Vissers et al., 2016*). Functional studies of products encoded by these genes are essential for our mechanistic understanding of ID pathogenesis.

Mutations causing ID were found in the gene encoding SLC6A17 (*Iqbal et al., 2015*; *Waltl, 2015*). SLC6A17, also known as Rxt1, NTT4, XT1, or B⁰AT3 (*el Mestikawy et al., 1994*; *Liu et al., 1993*), was discovered 30 years ago (*el Mestikawy et al., 1994*; *Liu et al., 1993*). It is predominantly expressed in the nervous system (*el Mestikawy et al., 1994*; *Fischer et al., 1999*; *Hägglund et al., 2013*; *Jursky and Nelson, 1999*; *Kachidian et al., 1999*; *Luque et al., 1996*; *Masson et al., 1995*; *Masson et al., 1999*). SLC6A17 protein was localized on synaptic vesicles (SVs) (*Fischer et al., 1999*; *Kachidian et al., 1999*; *Masson et al., 1999*). Those facts, together with its membership in the SLC6 or the neurotransmitter transporter (NTT) family, suggest that SLC6A17 could be a vesicular transporter for a neurotransmitter(s).

Early efforts failed to identify substrates transported by SLC6A17 (*Bröer, 2006*; *el Mestikawy et al., 1994*; *Liu et al., 1993*). A later study using pheochromocytoma (PC)12 cells and *SLC6A17*-tranfected Chinese hamster ovary (CHO) cells found that SLC6A17 could transport four amino acids (AA): proline (Pro), glycine (Gly), leucine (Leu), and alanine (Ala) (*Parra et al., 2008*). Another study using human embryonic kidney (HEK) cells reported transport of nine AAs: Leu, methionine (Met), Pro, cysteine (Cys), Ala, glutamine (Gln), serine (Ser), histidine (His), and Gly (*Zaia and Reimer, 2009*). There are three differences between these two studies: that the second study reported five more AAs (Met, Cys, Gln, Ser, and His) as substrates of SLC6A17, that the second study used modified SLC6A17 to facilitate its membrane localization (*Zaia and Reimer, 2009*), and that one reported Na⁺ dependence (*Zaia and Reimer, 2009*), whereas the other reported H⁺ dependence (*Parra et al., 2008*). However, none of these AAs have been found in the SVs. It remains unknown which of the four or nine AAs, if any, are present in the SVs in an SLC6A17-dependent manner in vivo.

Thus, how mutations in a single AA of SLC6A17 lead to ID is unknown, the substrate(s) physiologically transported by SLC6A17 is unknown, and behavioral phenotypes of animals lacking *Slc6a17* or carrying any mutation pathogenic in human patients have not been characterized.

We have now generated mouse mutants either lacking the *Slc6a17* gene or mimicking a point mutation found in ID patients. They had similar phenotypes including deficient learning and memory, indicating that the human mutation was a loss-of-function (LOF) mutation. These are the first animal models of ID caused by *Slc6a17* mutations. While two in vitro studies with cell lines suggested that SLC6A17 could transport up to nine neutral AAs, we could only find that Gln was present in the SVs. From multiple gain-of-function (GOF) and LOF experiments, we have obtained evidence that SLC6A17 is not only sufficient for Gln presence in the SVs in vivo, but it is also physiologically necessary for Gln presence in the SVs. Thus, we have found the endogenous substrate for SLC6A17. Of the two known human mutations pathogenic for ID (*Iqbal et al., 2015*), we provide evidence that one caused the SLC6A17 to be mislocalized subcellularly and the other, while still on SVs, was defective in Gln transport. Decreases in Gln in SVs caused by SLC6A17 mutations were not correlated with any decrease in glutamate (Glu) or gamma-aminobutyric acid (GABA), dissociating vesicular Gln from the Glu/GABA-Gln cycle between neurons and glia. In addition to dissecting the molecular mechanisms underlying ID caused by *Slc6a17* mutations, we report for the first time that Gln is robustly and reproducibly detected in the SVs, which should stimulate further research into its potential roles in neurotransmission.

## Results

### Pattern of *Slc6a17* expression

To examine the pattern of *Slc6a17* expression, we designed a knock-in mouse line *Slc6a17*[-2A-CreERT2] (*Figure 1A*). These mice were generated by in-frame fusion of a T2A sequence (*Ahier and Jarriault, 2014*; *Daniels et al., 2014*; *Trichas et al., 2008*) and CreERT2 (*Feil et al., 1996*; *Gu et al., 1994*; *Indra et al., 1999*; *Sauer and Henderson, 1988*) to the C terminus of *Slc6a17* with its stop codon removed

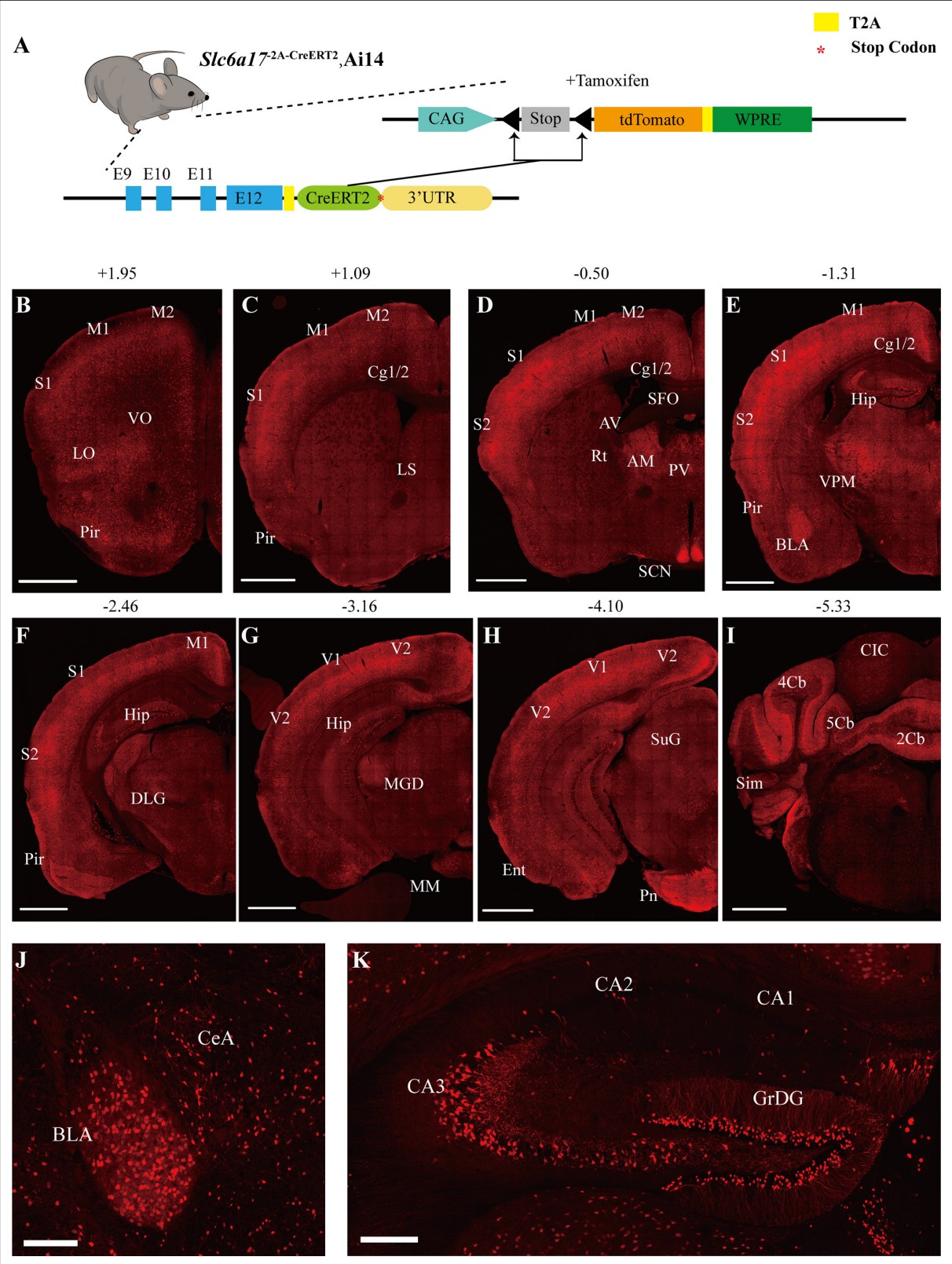

**Figure 1.** *Slc6a17* expression in the mouse brain. (**A**) A schematic diagram illustrating the strategy for the generation of *Slc6a17*-2A-CreERT2 mice. Crossing of *Slc6a17*-2A-CreERT2 mice with Ai14 (LSL-tdTomato) mice allowed specific labeling of *Slc6a17*-expressing neuron after tamoxifen injection. More details in *Figure 1—figure supplement 1A*. (**B–I**) Representative coronal sections of *Slc6a17*-2A-CreERT2::Ai14 mice. Numbers above images indicate the anteroposterior position of the section from Bregma in millimeters (mm), based on *Paxinos and Franklin, 2019*. Scale bars = 1 mm. (**J**) *Slc6a17*-positive

*Figure 1 continued on next page*

*Figure 1 continued*

neurons in the basolateral amygdaloid nucleus (BLA) and the central amygdaloid nucleus (CeA). Scale bars = 100 µm. (**K**) Hippocampal expression of *Slc6a17*. *Slc6a17*-positive neurons are densely distributed in the granule cell layer of the dentate gyrus (GrDG), and CA3, with little expression in the CA1 and the CA2. Scale bars = 200 µm. Abbreviations: 2Cb, lobule 2 of the cerebellar vermis; 3Cb, lobule 3 of the cerebellar vermis; 4/5Cb, lobule 4 and 5 of the cerebellar vermis; AON, accessory olfactory nucleus; AM, anteromedial thalamic nucleus; AV, anteroventral thalamic nucleus; BLA, basolateral amygdaloid nucleus; Ent, entorhinal cortex; CB, cerebellum; Cg1/2, cingulate ccortex; CIC, central nucleus of the inferior colliculus; Ctx, cortex; Hip, hippocampus; LO, lateral orbital cortex; LS, lateral septal nucleus; M1, primary motor cortex; M2, secondary motor cortex; OB, main olfactory bulb; Pir, piriform cortex; Pn, pontine nuclei; Rt, reticular thalamic nucleus; S1, primary somatosensory cortex; S2, secondary somatosensory cortex; SCN, suprachiasmatic nucleus; SFO, subfornical organ; SuG, superficial gray layer of superior colliculus; Sim, simple lobule; Th, thalamus; V1, primary visual cortex; V2, secondary visual cortex; VPM, ventral posteromedial nucleus.

The online version of this article includes the following figure supplement(s) for figure 1:

**Figure supplement 1.** Generation of *Slc6a17* knock-in mice.

(*Figure 1—figure supplement 1A*). We crossed the *Slc6a17*[-2A-CreERT2] mice with the Ai14 reporter line, which contained floxed stop-tdTomato (*Madisen et al., 2010*). Treatment with tamoxifen removed the stop codon and thus allowed specific expression of tdTomato in *Slc6a17*-positive cells (*Figure 1A*, *Figure 1—figure supplement 1A*). Labeled cells were exclusively neuronal with no glial expression detected.

Slc6a17 expression was found in the neocortex, the thalamus, the amygdala, the hippocampus, the pontine nuclei, and the brainstem (*Figure 1B–K*, *Figure 1—figure supplement 1B, C*, *Supplementary file 2*). Strong expression of *Slc6a17* was detected in the dentate gyrus (DG) and the CA3 region of the hippocampus (*Figure 1K*), which is essential for spatial learning and memory. Little expression was observed in either CA1 or CA2 (*Figure 1K*). *Slc6a17* was also expressed in the basolateral amygdala (BLA) (*Figure 1J*), a brain area essential for emotion and fear learning.

## Behavioral deficits of *Slc6a17* mutant mice

To investigate the functional role of *Slc6a17*, we generated a knock-out (KO) mouse mutant line with exon 2 deleted from the *Slc6a17* gene (*Figure 2A*, *Figure 3—figure supplement 1A*). These *Slc6a17*-KO mice were not significantly different from the wild type (WT, *Slc6a17*[+/+]) mice in body weight (*Figure 2—figure supplement 1A*), basal activities (*Figure 2—figure supplement 1B*), or short-term memory (*Figure 2—figure supplement 1C*).

To investigate learning and memory of *Slc6a17*-KO mice, we first tested *Slc6a17*-KO mice on a classical associative learning model of contextual and cued fear (*Hitti and Siegelbaum, 2014*; *Phillips and LeDoux, 1992*; *Figure 2B*). Both the WT and the heterozygous (*Slc6a17*[+/-]) mice spent a significant amount of time freezing (*Figure 2C*) on day 2 in Context-a environment. However, homozygous (*Slc6a17*[-/-]) mutants showed a dramatic decrease in freezing behavior compared to the other genotypes (*Figure 2C*). On day 3, both *Slc6a17*[+/+] and *Slc6a17*[+/-] mice showed significantly increased freezing behavior (*Figure 2C*), whereas *Slc6a17*[-/-] mutants showed significant memory impairment in response to the tone (*Figure 2C*). These results indicate that the *Slc6a17*-KO mice were severely deficient in contextual and cued fear memory.

We then tested hippocampus-dependent spatial learning and memory by evaluating the performance of mice in the Morris water maze (*Morris, 1984*; *Morris et al., 1982*; *Figure 2D*). During training, control (*Slc6a17*[+/+] and *Slc6a17*[+/-]) mice showed a gradually decreasing latency to find the platform in both visible and hidden sessions (*Figure 2E*), whereas *Slc6a17*[-/-] mutants exhibited significantly longer latency to find the platform (*Figure 2E*). Representative swimming traces of each genotype in probe trial are shown in *Figure 2F*. During the probe trial, control mice spent more time in the target quadrant (*Figure 2F*), whereas *Slc6a17*[-/-] mutants spent much less time searching the platform quadrant (*Figure 2G and H*). The frequency that control animals crossed through the platform area was above chance level (*Figure 2G and H*), whereas *Slc6a17*[-/-] mutants crossed the platform area much less than the other genotypes (*Figure 2G and H*).

Taken together, these data indicate that *Slc6a17*-KO mice are significantly impaired both in contextual and cued fear memory and in hippocampus-dependent spatial learning and memory.

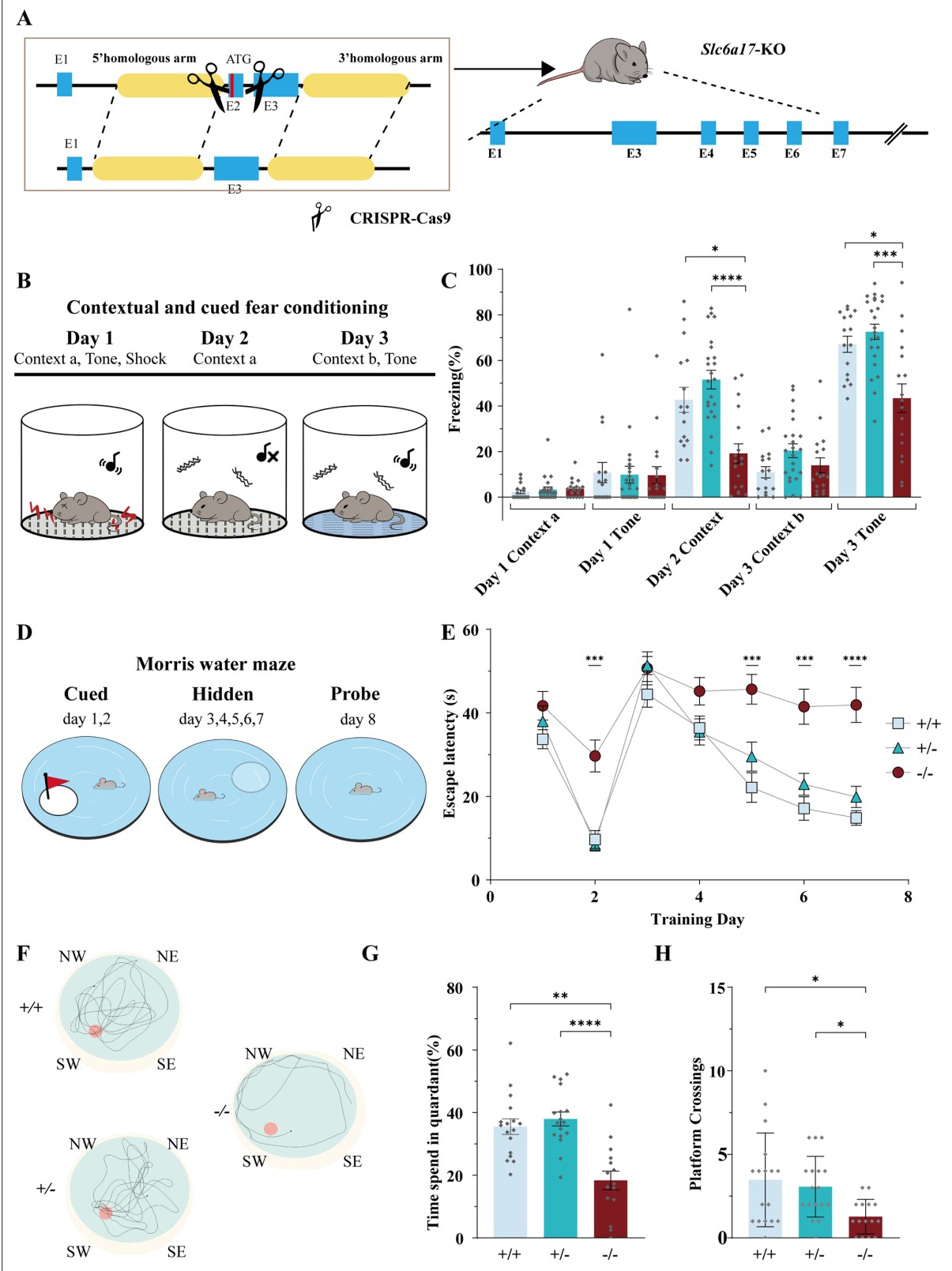

**Figure 2.** *Slc6a17*-KO mice exhibited impaired memory formation. (**A**) A schematic diagram illustrating the knock-out strategy for generating *Slc6a17*-KO mice by CRISPR/Cas9. Exon 2 encoding the first 99 amino acids of *Slc6a17* was deleted. More details in *Figure 3—figure supplement 1A*. (**B**) A diagram of experiments for contextual and cued fear conditioning (n = 16, 23, and 18 for *Slc6a17^{+/+}*, *Slc6a17^{+/-}*, and *Slc6a17^{-/-}*, respectively). Detailed description in 'Experimental procedures.' (**C**) *Slc6a17^{-/-}* mice exhibited a significant decrease in freezing behavior in both cued and contextual fear

*Figure 2 continued*

conditioning (day 2 context a, p=0.0062 for *Slc6a17*[+/+] vs. *Slc6a17*[-/-], p<0.0001 for *Slc6a17*[+/-] vs. *Slc6a17*[-/-]; Day 3 tone, p=0.0093 for *Slc6a17*[+/+] vs. *Slc6a17*[-/-], p=0.0012 for *Slc6a17*[+/-] vs. *Slc6a17*[-/-]). (**D**) A diagram of experiments for modified Morris water maze task (n = 15, 18, and 16 for *Slc6a17*[+/+], *Slc6a17*[+/-], and *Slc6a17*[-/-], respectively). Detailed description in 'Experimental procedures.' (**E**) Latency to find the hidden platform during the training session. *Slc6a17*[-/-] mice differed significantly from *Slc6a17*[+/+] and *Slc6a17*[+/-] mice (two-way ANOVA; main effect of training day: $F_{(4.681, 210.7)}$ = 46.55, p<0.0001; main effect of genotype: $F_{(2, 45)}$ = 21.53, p<0.0001; main effect of training day × genotype: $F_{(12, 270)}$ = 3.762, p<0.0001; Tukey's multiple comparisons test: day 2, p=0.0004 for *Slc6a17*[+/+] vs. *Slc6a17*[-/-], p=0.0002 for *Slc6a17*[+/-] vs. *Slc6a17*[-/-]; day 5, p=0.0002 for *Slc6a17*[+/+] vs. *Slc6a17*[-/-], p=0.008 for *Slc6a17*[+/-] vs. *Slc6a17*[-/-]; day 6, p=0.0002 for *Slc6a17*[+/+] vs. *Slc6a17*[-/-], p=0.0027 for *Slc6a17*[+/-] vs. *Slc6a17*[-/-]; day 7, p<0.0001 for *Slc6a17*[+/+] vs. *Slc6a17*[-/-], p=0.0005 for *Slc6a17*[+/-] vs. *Slc6a17*[-/-]). (**F**) Representative swimming traces of each genotype. (**G**) Percentage of time spent in the target quadrant was significantly different (p=0.0004 for *Slc6a17*[+/+] vs. *Slc6a17*[-/-], p<0.0001 for *Slc6a17*[+/-] vs. *Slc6a17*[-/-]). (**H**) Number of crossings through the platform area was significantly different (p=0.0194 for *Slc6a17*[+/+] vs. *Slc6a17*[-/-], p=0.0054 for *Slc6a17*[+/-] vs. *Slc6a17*[-/-]). Here and hereafter, data in the figure are presented as the mean ± SEM, with * indicating p<0.05, ** indicating p<0.01, *** indicating p<0.001, and **** indicating p<0.0001.

The online version of this article includes the following source data and figure supplement(s) for figure 2:

**Source data 1.** Data points for *Figure 2C, E, G, and H*.

**Figure supplement 1.** Phenotypical analysis of *Slc6a17*-KO mice.

**Figure supplement 1—source data 1.** Data points for *Figure 2—figure supplement 1A–F*.

**Figure supplement 2.** Mass spectrometry (MS) analysis of *Slc6a17* KO mice.

**Figure supplement 2—source data 1.** Data points for *Figure 2—figure supplement 2*.

## Behavioral deficits in *Slc6a17*[P663R] mutant mice

We next generated a mouse line carrying a point mutation (P633R) found to be pathogenic in ID patients (*Iqbal et al., 2015*). Three repeats of the hemagglutinin (HA) epitope (*Kolodziej and Young, 1991*; ) and three repeats of the FLAG tag were fused in-frame to the C terminus of the endogenous P633R mutant (*Figure 3A*, *Figure 3—figure supplement 1B*). Homozygous (*Slc6a17*[P633R/P633R]) mutant mice were not significantly different from the WT (*Slc6a17*[+/+]) or heterozygous (*Slc6a17*[P633R/+]) in body weight (*Figure 3—figure supplement 2A*), basal activities (*Figure 3—figure supplement 2B*), or novel objection recognition (*Figure 3—figure supplement 2C*).

We then investigated learning and memory in *Slc6a17*[P633R] mice. In contextual and cued fear conditioning (*Figure 3C*), *Slc6a17*[P633R/P633R] mutants showed significantly less freezing behavior compared to *Slc6a17*[+/+] and heterozygous (*Slc6a17*[P633R/+]) mice on day 2 in Context-a environment (*Figure 3D*). On day 3, *Slc6a17*[P633R/P633R] mutants showed a lower level of freezing behavior induced by cued tone compared to *Slc6a17*[+/+] and *Slc6a17*[P633R/+] mice (*Figure 3D*). These results indicate that *Slc6a17*[P633R] mice are defective in contextual and cued fear memory.

We then tested *Slc6a17*[P633R] mice in the Morris water maze for hippocampal spatial learning and memory (*Figure 3E*; *Morris, 1984*; *Morris et al., 1982*). During training, *Slc6a17*[+/+] and *Slc6a17*[P633R/+] mice had a similar pattern in their learning curves with gradual decreases in the latency to find the platform in two sessions (*Figure 3F*). Similar to the *Slc6a17*-KO mice, *Slc6a17*[P633R/P633R] mutants required significantly longer time to find the platform in both visible and hidden sessions (*Figure 3F*). Representative swimming traces of each genotype in probe trials are shown in *Figure 3G*. During the probe trial, *Slc6a17*[+/+] and *Slc6a17*[P633R/+] mice spent more time in the target quadrant (*Figure 3H*) and had more crossings through the platform area than *Slc6a17*[P633R/P633R] mutants (*Figure 3I*).

Taken together, these data indicate that *Slc6a17*[P633R] mice are also significantly impaired both in contextual and cued fear memory and in hippocampus-dependent long-term spatial learning. Because all the phenotypes of *Slc6a17*[P633R] are similar to Slc6a7-KO, the point mutation P633R is a LOF mutation.

## Vesicular localization of SLC6A17

Previous studies with anti-SLC6A17 antibodies provided evidence that SLC6A17 was localized in SVs (*Fischer et al., 1999*; *Kachidian et al., 1999*; *Masson et al., 1999*). We have now used multiple approaches to analyze the subcellular localization of SLC6A17.

*Slc6a17*[-HA-2A-iCre] mice were generated with the C terminus of the endogenous SLC6A17 protein fused in frame with three repeats of the HA epitope (*Figure 4A*, *Figure 4—figure supplement 1A*). We used differential centrifugation to purify SVs from the brains of *Slc6a17*[-HA-2A-iCre] mice (*Evans, 2015*; *Huttner et al., 1983*). As shown in *Figure 4—figure supplement 1C*, SLC6A17-HA was enriched in

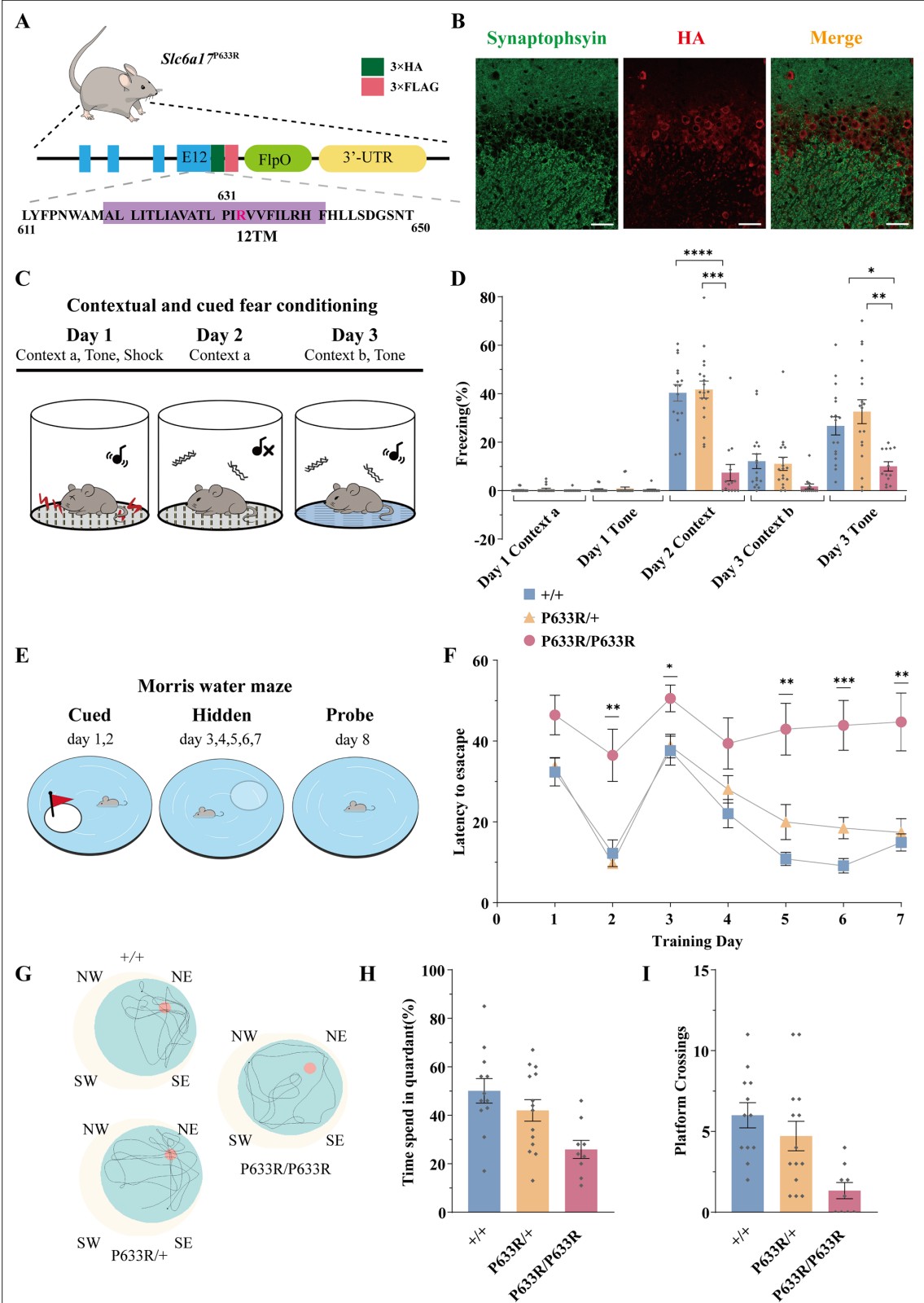

**Figure 3.** Deficient memory in *Slc6a17*[P633R] mice. (**A**) A diagram illustrating the knock-in strategy for *Slc6a17* with a pathogenic point mutation (*Slc6a17*[P633R]). It was tagged with three repeats of the HA epitope and three repeats of the FLAG epitope. More details in *Figure 3—figure supplement 1B*. (**B**) Immunocytochemistry of *Slc6a17*[P633R] mouse brains with the anti-HA antibody showed that the subcellular localization of SLC6A17[P633R]-HA did not co-localized with synaptophsin. It was present in the cytoplasm. Representative view of CA3 is shown. Scale bar = 50 µm. (**C**) Experiments

*Figure 3 continued on next page*

*Figure 3 continued*

for contextual and cued fear conditioning (n = 17, 18, and 14 for *Slc6a17*[+/+], *Slc6a17*[P633R/+], and *Slc6a17*[P633R/P633R], respectively). Detailed description in 'Experimental procedures.' (**D**) *Slc6a17*[P633R] homozygous mutants exhibited significant decrease of freezing behavior in both cued and contextual fear conditioning (day 2 context A, p<0.0001 for *Slc6a17*[+/+] vs. *Slc6a17*[P633R/P633R], p<0.0001 for *Slc6a17*[P633R/+] vs. *Slc6a17*[P633R/P633R]; Day 3 tone, p=0.0017 for *Slc6a17*[+/+] vs. *Slc6a17*[P633R/P633R], p=0.0009 for *Slc6a17*[P633R/+] vs. *Slc6a17*[P633R/P633R]). (**E**) Cartoon figure of experiments for Morris water maze task (n = 12, 15, and 9 for *Slc6a17*[+/+], *Slc6a17*[P633R/+], and *Slc6a17*[P633R/P633R], respectively). Detailed description in 'Experimental procedures.' (**F**) Latency to find the hidden platform during the training session. *Slc6a17*[P633R/P633R] mice showed significantly different learning curve from *Slc6a17*[+/+] and *Slc6a17*[P633R/+] mice (two-way ANOVA; main effect of training day: F (4.381, 144.6) = 20.02, p<0.0001; main effect of genotype: F (2, 33) = 22.65, p<0.0001; main effect of training day × genotype: F (12, 198) = 2.389, p=0.0067; Tukey's multiple-comparisons test: day 2, p=0.0150 for *Slc6a17*[+/+] vs. *Slc6a17*[P633R/P633R], p=0.0078 for *Slc6a17*[P633R/+] vs. *Slc6a17*[P633R/P633R]; day 3, p=0.0403 for *Slc6a17*[+/+] vs. *Slc6a17*[P633R/P633R], p=0.0388 for *Slc6a17*[P633R/+] vs. *Slc6a17*[P633R/P633R]; day 5, p=0.0023 for *Slc6a17*[+/+] vs. *Slc6a17*[P633R/P633R], p=0.0243 for *Slc6a17*[P633R/+] vs. *Slc6a17*[P633R/P633R]; day 6, p=0.0199 for *Slc6a17*[+/+] vs. *Slc6a17*[P633R/+], p=0.0010 for *Slc6a17*[+/+] vs. *Slc6a17*[P633R/P633R], p=0.0077 for *Slc6a17*[P633R/+] vs. *Slc6a17*[P633R/P633R]; day 7, p=0.0074 for *Slc6a17*[+/+] vs. *Slc6a17*[P633R/P633R], p=0.0129 for *Slc6a17*[P633R/+] vs. *Slc6a17*[P633R/P633R]). (**G**) Representative swimming traces of each genotype. (**H**) Percentage of time spent in the target quadrant was significantly different (p=0.0031 for *Slc6a17*[+/+] vs. *Slc6a17*[P633R/P633R], p=0.0325 for *Slc6a17*[P633R/+] vs. *Slc6a17*[P633R/P633R]). (**I**) Number of crossings through the platform area was significantly different (p=0.0003 for *Slc6a17*[+/+] vs. *Slc6a17*[P633R/P633R], p=0.0123 for *Slc6a17*[P633R/+] vs. *Slc6a17*[P633R/P633R]). Data in the figure are presented as the mean ± SEM, with * indicating p<0.05, ** indicating p<0.01, *** indicating p<0.001, and **** indicating p<0.0001.

The online version of this article includes the following source data and figure supplement(s) for figure 3:

**Source data 1.** Data points for *Figure 3D, H, and I*.

**Figure supplement 1.** Generation of *Slc6a17* mutant mice.

**Figure supplement 1—source data 1.** Original files of the full raw unedited blots for *Figure 3—figure supplement 1D*.

**Figure supplement 1—source data 2.** Uncropped blots with the relevant bands lablled for *Figure 3—figure supplement 1D*.

**Figure supplement 2.** Phenotypical analysis of *Slc6a17*[P633R] mice.

**Figure supplement 2—source data 1.** Data points for *Figure 3—figure supplement 2A–F*.

LP2 fraction, similar to the other SV proteins such as synaptotagmin 1 (Syt1) (*Geppert et al., 1994*), vesicular ATPase (V-ATPase) (*Cidon and Sihra, 1989*), and synaptobrevin 2 (Syb2) (*Link et al., 1992*; *Schiavo et al., 1992*), but different from postsynaptic proteins such as postsynaptic density 95 (PSD95) (*Cho et al., 1992*; *Woods and Bryant, 1991*). Sucrose gradient centrifugation of the LP2 fraction showed copurification of SLC6A17-HA with SV proteins including synaptophysin (Syp) (*Jahn et al., 1985*; *Leube et al., 1987*; *Wiedenmann and Franke, 1985*), Syt1, and VGluT1 (*Bellocchio et al., 2000*; *Takamori et al., 2000*), but not with proteins localized to the postsynaptic membrane (PSD95) or (SNAP23) (*Ravichandran et al., 1996*; *Suh et al., 2010*), the endoplasmic reticulum (ER) (ERp72), the endosome (EEA1) (*Mu et al., 1995*), and the proteosome (PSMC6) or the trans-Golgi network (synaptaxin 6, STX6) (*Figure 4D*).

We used the anti-HA magnetic beads to immunoisolate SLC6A17-HA protein from *Slc6a17*[-HA-2A-iCre] mice (*Slc6a17*[HA/+]), with WT (*Slc6a17*[+/+]) mice as a control (*Figure 4C*, *Figure 4—figure supplement 1D*). After examining markers of subcellular organelles, the fraction containing SLC6A17-HA was found to be co-localized with SV markers such as Syp, Syt1, Syb2, VATPase, and vesicular neurotransmitter transporters including VGluT1, VGluT2, and VGAT (*Figure 4C*, *Figure 1—figure supplement 1D*), but not with markers of axons or dendrites (GLUT4 or transferrin receptor), ER (ERp72), the lysosome (LAMP2, LC3b, or cathepsin D), the cytoplasmic membrane (SNAP23, PSD95, and GluN1), the Golgi apparatus (GM130 and Goglin-97), mitochondria (VADC), the active zone (ERC1b/2), the endosome (EEA1), and the proteosome (PSMC6) (*Figure 4C*, *Figure 1—figure supplement 1D*). In addition, immunostaining of HA-tagged SLC6A17 in *Slc6a17*[HA/+] mouse showed puncta co-localized with the anti-Syp positive immunoreactivity (*Figure 4B*, *Figure 3—figure supplement 1C and B*).

We also used genetically assisted EM to confirm SLC6A17 localization. APEX2 is derived from ascorbate peroxidase (APEX) and can be used to genetically label proteins for EM inspection (*Lam et al., 2015*; *Martell et al., 2017*). We fused APEX2 in-frame to the C terminus of SLC6A17 (*Figure 4E*). SLC6A17-APEX2 fusion protein was delivered into the adult mouse brain via systemic administration of a BBB-crossing adeno-associated virus (AAV9) variant AAV-PHP.eb, under human synapsin promoter (hSyn) (*Chan et al., 2017*; *Deverman et al., 2016*; *Figure 4E*). SLC6A17-APEX2 was indeed localized on the SVs (*Figure 4F*, *Figure 4—figure supplement 2A, C, E, and G*). Quantitative analysis showed distinct populations of SVs distinguished by electron density (*Figure 4F*, *Figure 4—figure supplement 2B, D, F, and H*).

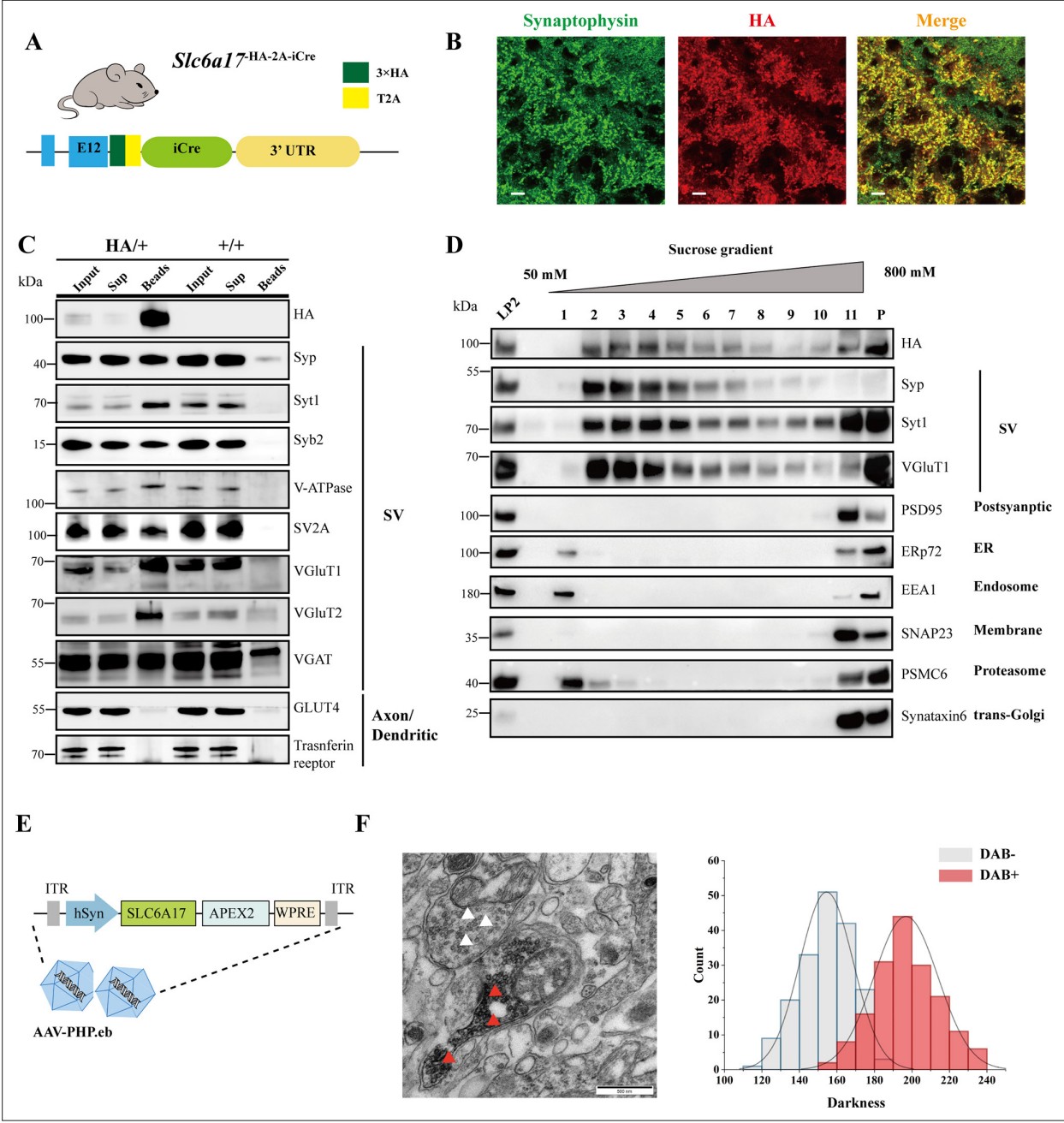

**Figure 4.** Biochemical and genetically assisted electron microscopy (EM) validation of the vesicular localization of SLC6A17. (**A**) A schematic diagram illustrating the knock-in strategy for generating *Slc6a17*-HA-2A-iCre mice. More details in *Figure 1—figure supplement 1D*. (**B**) Higher magnification views of co-immunostaining by anti-Syp and anti-HA antibodies in the hippocampus. Scale bar = 10 µm. (**C**) Immunoisolation of SLC6A17-HA containing vesicles by the anti-HA antibody coated on magnetic beads. SLC6A17-HA fraction was positive for Syp, Syt1, Syb2, v-ATPase, VGluT1, VGluT2, and vGAT, but negative for GluT4 and transferrin receptor. (**D**) Further purification of the LP2 fraction by sucrose gradient showed that SLC6A17-HA was co-immunoisolated with Syp, Syt1, and VGluT1, but not PSD95, ERp72, EEA1, SNAP23, PSMC6, or STX6. (**E**) A schematic diagram illustrating the APEX2-based labeling strategy with AAV-PHP.eb virus mediated SLC6A17-APEX2 overexpression in vivo. SLC6A17 was fused in-frame to three repeats of the HA tag, a V5 tag, and APEX2. (**F**) Representative EM image of synaptic vesicles (SVs) labeled by SLC6A17-APEX2 and darkness distribution of DAB-positive and DAB-negative SVs in sections of *Slc6a17*-APEX2 mouse brains. Red arrow pointing to APEX2 labeled SVs. White arrow pointing to unlabeled SVs.

The online version of this article includes the following source data and figure supplement(s) for figure 4:

**Source data 1.** Original files of the full raw unedited blots for *Figure 4C and D*.

**Source data 2.** Uncropped blots with the relevant bands labeled for *Figure 4C and D*.

*Figure 4 continued on next page*

**Figure supplement 1.** Immunoisolation and immunohistochemistry of *Slc6a17*-HA mice.

**Figure supplement 2.** Electron microscopy (EM) confirmation of the synaptic vesicle (SV) localization of SLC6A17-APEX2.

Our results from biochemistry and genetically assisted EM analysis support the SV localization of SLC6A17.

## Functional significance of the SV localization of SLC6A17

We used a monoclonal anti-Syp antibody to immunoisolate SVs from brains of $Slc6a17^{+/+}$ and $Slc6a17^{-/-}$ mice (*Boyken et al., 2013*; *Grønborg et al., 2010*; *Jahn et al., 1985*). Quantitative proteomic analysis showed that only SLC6A17 was dramatically reduced in SVs immunopurified from *Slc6a17*-KO mice, whereas other transporters such as VGluT1, VGluT2, VGluT3, VGAT, VMAT2, SV2A, SV2B, SV2C, ZnT3, and VAT-1 were not different between $Slc6a17^{-/-}$ and $Slc6a17^{+/+}$ (*Figure 2—figure supplement 2*). Thus, *Slc6a17* gene knockout indeed specifically decreased SLC6A17 on SVs, but none of the other transporters.

To investigate the functional significance of the subcellular localization of SLC6A17, we examined the subcellular localization of SLC6A17$^{P633R}$ protein in the mouse brain. SLC6A17$^{P633R}$ mutation was found in human ID patients (*Iqbal et al., 2015*) and transfection of a cDNA encoding this mutant into cultured primary neurons from mice found that its distribution was changed from that of the WT SLC6A17 (*Iqbal et al., 2015*), but the disturbance on SV localization was unambiguous and the effect of the mutation on the endogenous protein was unknown.

With the endogenous SLC6A17$^{P633R}$ protein tagged by the HA and FLAG epitopes (*Figure 3A*, *Figure 3—figure supplement 1B*), we found that SLC6A17$^{P633R}$-HA signal was primarily localized to the soma with no overlap detected with the anti-Syp antibody staining (*Figure 3B*, *Figure 3—figure supplement 1C*). We also performed differential centrifugation and sucrose gradient purification. HA and FLAG signals were presented in LP2 but did not co-purified with SVs in sucrose gradients, indicated that the SLC6A17$^{P633R}$ protein was not present in SVs (*Figure 3—figure supplement 1D*).

## Presence of Gln in immunoisolated SVs

Neurotransmitters such as Glu and GABA have been reliably found in SVs immunoisolated from the mammalian brain (*Bradberry et al., 2022*; *Burger et al., 1991*; *Burger et al., 1989*; *Chantranupong et al., 2020*; *Martineau et al., 2013*). Previous experiments suggest that SLC6A17 could transport Ala, Gly, Leu, and Pro into PC12 and CHO cells (*Parra et al., 2008*), or Ala, Gly, Leu, Pro, Cys, Gln, Gly, His, and Ser into HEK cells (*Zaia and Reimer, 2009*). However, none of the reported substrates of SLC6A17 has been reliably detected in SVs.

To analyze the contents of SVs, we used the anti-Syp antibody to immunoisolate SVs from the mouse brain (*Burger et al., 1991*; *Burger et al., 1989*; *Martineau et al., 2013*), before their contents were subjected to chemical analysis by LC-MS (*Figure 5A and B*). IgG was used as a control for the anti-Syp antibody in SV immunoisolation. The specificity of the anti-Syp antibody-mediated SVs immunoisolation was confirmed (*Figure 5—figure supplement 1A*), with a total of 16 markers for SVs, lysosomes, endosomes, the Golgi apparatus, mitochodria, the ER, and pre- or postsynaptic membrane.

The ratio of a molecule immunoisolated with the anti-Syp antibody to that with IgG was calculated and analyzed (*Figure 5C and D*). Glu, GABA, and ACh were reliably detected from SVs immunoisolated with the anti-Syp antibody (*Figure 5C–G*, *Figure 5—figure supplement 1B, D, and F*). Monoamine neurotransmitters such as 5-hydroxytryptamine (5-HT) (*Figure 5—figure supplement 1C*), dopamine (DA) (*Figure 5—figure supplement 1G*), and histamine (*Figure 5—figure supplement 1E*) could also be detected but not further analyzed for this paper. Hereafter, we focused our analysis on Glu, GABA, and ACh as the positive controls and the nine previously reported AAs as candidate substrates of SLC6A17 in vivo.

Surprisingly, of the nine AAs reported previously using cultured cells, eight (Ala, Gly, Leu, Pro, Cys, Gly, His, and Ser) were not found to be significantly enriched in SVs (*Figure 5C and D*). Only Gln was reproducibly found to be enriched in the anti-Syp antibody immunoisolated SVs (*Figure 5C, D and H*).

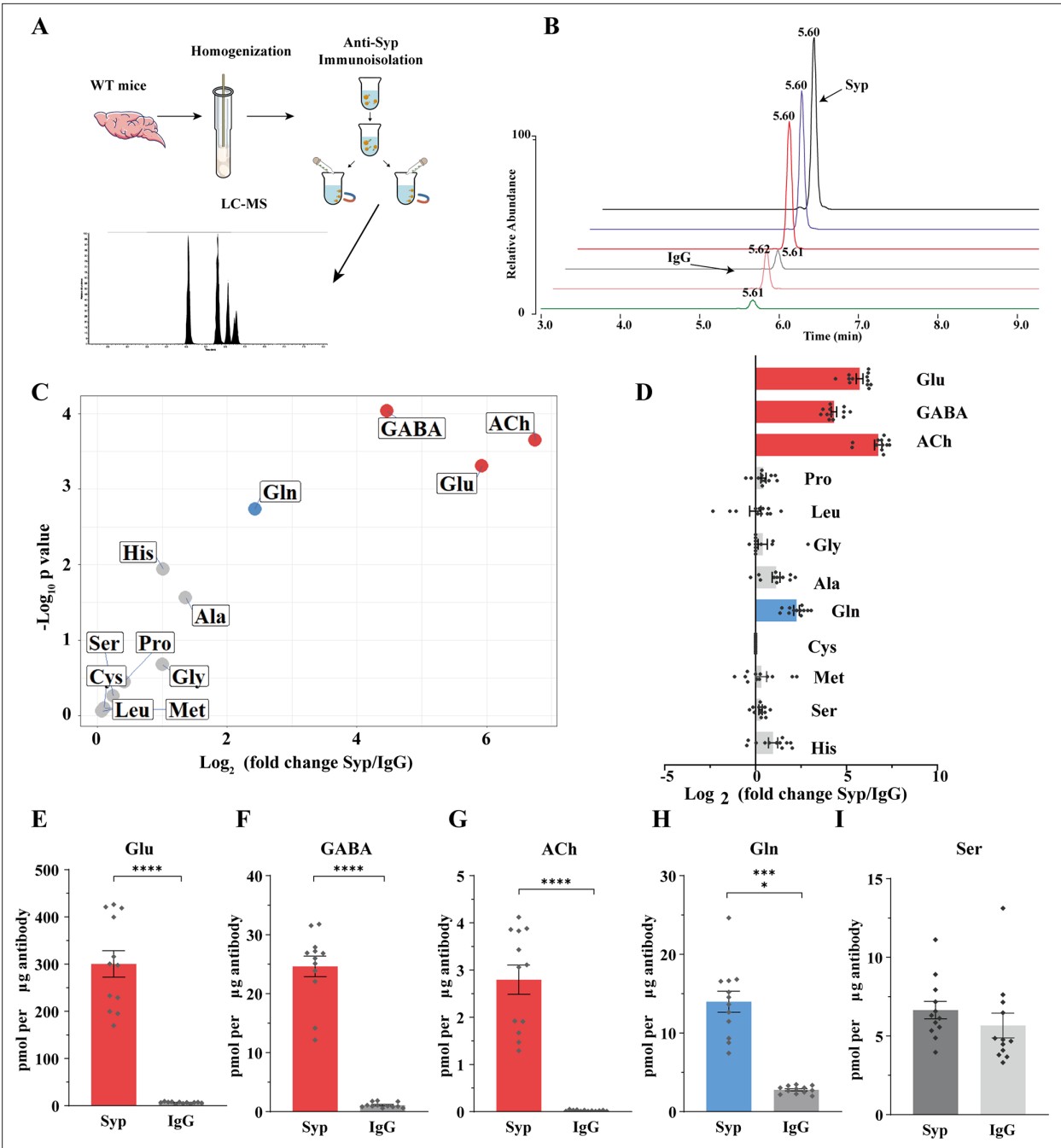

**Figure 5.** Presence of glutamine in synaptic vesicles (SVs) from the mouse brain. (**A**) A schematic diagram illustrating the procedure to immunoisolate SVs with the anti-Syp antibody for liquid chromatography coupled to mass spectrometry (LC-MS) analysis of the contents in the SVs. (**B**) Representative MS results showing Gln signals from SVs immunoisolated by the anti-Syp antibody (anti-Syp) vs. the control sample immunoisolated with IgG. (**C**) Volcano plot of chemical contents in the SVs isolated by anti-Syp vs. IgG. The y axis shows p-values in $\log_{10}$ and the x axis shows the $\log_2$ of the ratio of the level of a molecule immunoisolated by anti-Syp vs. IgG. Classical neurotransmitters Glu, GABA, and ACh, as well as previously reported substrates of SLC6A17 are listed. (**D**) Ratios of the level of a chemical immunoisolated by anti-Syp vs. IgG (transformed into $\log_2$). (**E–I**) Chemicals were quantified to mole per μg antibody (n = 12 for each group from four different animals with three replicates each): Glu (**E**, p<0.0001 for anti-Syp vs. IgG); GABA (**F**, p<0.0001 for anti-Syp vs. IgG); ACh (**G**, p<0.0001 for anti-Syp vs. IgG); Gln (**H**, p<0.0001 for anti-Syp vs. IgG); Ser (**I**, p=0.7553 for anti-Syp vs. IgG).

The online version of this article includes the following source data and figure supplement(s) for figure 5:

**Source data 1.** Data points for *Figure 5C–I*.

**Figure supplement 1.** Liquid chromatography coupled to mass spectrometry (LC-MS) analysis of SVs immunoisolated by the anti-Syp antibody.

*Figure 5 continued on next page*

*Figure 5 continued*

**Figure supplement 1—source data 1.** Original files of the full raw unedited blots for *Figure 5—figure supplement 1A*.

**Figure supplement 1—source data 2.** Uncropped blots with the relevant bands labeled for *Figure 5—figure supplement 1A*.

There could be several reasons why molecules transported into cultured cells were not detected in SVs of the mouse brain, including, for example, relative abundance in subsets of SVs specific for different transporters, or the redundancy of multiple transporters for a single molecule among the nine candidates. However, the positive finding of Gln in the SVs is definitive.

## Enrichment of Gln in SLC6A17 containing SVs from the mouse brain

The above results with SVs purified by the anti-Syp antibody revealed Gln presence in SVs, but did not show association of Gln with specific transporter on SV, because Syp is a universal marker of SVs (*Jahn et al., 1985*; *Wiedenmann and Franke, 1985*; *Leube et al., 1987*; *Südhof et al., 1987*). To determine the potential relationship of SLC6A17 with Gln in the SVs, we used magnetic beads coated with the anti-HA antibody to immunoisolate SLC6A17-positive SVs from *Slc6a17*[-HA-2A-iCre] mice (*Figure 4C*, *Figure 4—figure supplement 1D*).

The anti-HA beads could enrich SVs containing SLC6A17 from the brains of *Slc6a17*[-HA-2A-iCre] mice but not from wt mice, as confirmed by the analysis of 23 markers (*Figure 4C*, *Figure 4—figure supplement 1D*). SVs thus isolated (*Figure 6A*) were subject to chemical analysis with LC-MS (*Figure 6A and B*). Glu, GABA, and ACh were detected in SVs containing SLC6A17-HA (*Figure 6C–G*), consistent with the immunoblot results that these SVs contained VGluT1, VGluT2, and VGAT (*Figure 4C*). Significant enrichment of Gln was detected in SVs containing SLC6A17-HA (*Figure 6C, D, and H*). By contrast, the other eight AAs had not been found to be enriched in SVs containing SLC6A17-HA (*Figure 6C, D, and I*).

Thus, only Gln has been reproducibly found to be enriched in SLC6A17 containing SVs in vivo.

## Sufficiency of SLC6A17 for Gln localization in SVs

To further investigate whether SLC6A17 could increase Gln in SVs, we overexpressed SLC6A17 in the mouse brain. AAV-PHP.eb was systematic administrated to express SLC6A17-HA in the mouse brain (OE-SLC6A17-HA) under hSyn promoter (*Chan et al., 2017*; *Deverman et al., 2016*; *Figure 7A*). The overexpressed SLC6A17-HA protein was localized in SVs, as analyzed by immunoblot (*Figure 7—figure supplement 1A*).

We then analyzed the chemical contents of immunoisolated SVs by LC-MS. The ratio of a molecule purified from OE-SLC6A17-HA mice vs. that purified from WT mice was calculated and analyzed (*Figure 7B*). SVs from mice overexpressing SLC6A17-HA contained significantly higher levels of Glu and GABA (*Figure 7B–G*). Gln was dramatically increased (*Figure 7B and F*), to the extent that Gln was higher than GABA in SVs overexpressing SLC6A17-HA (*Figure 7B, D, F, K, and M*). Quantification of the enriched molecules in SVs containing SLC6A17-HA showed Gln to be approximately two times that of GABA (*Figure 7D and F*). In addition, His was moderately increased in SVs overexpressing SLC6A17-HA (*Figure 7B*, *Figure 7—figure supplement 1C, D*). These results from our virally introduced overexpression experiments support that SLC6A17 is sufficient for Gln transport into SVs in vivo.

## Functional significance of SLC6A17 for Gln presence in the SVs

*SLC6A17*[G162R] is another pathogenic mutation of human ID (*Iqbal et al., 2015*), with a point mutation in the third transmembrane domain of SLC6A17. To investigate whether SLC6A17[G162R] affected Gln transportation, we used AAV-PHP.eb to overexpress either SLC6A17[G162R]-HA or Syp-HA in mouse brains under hSyn promoter. Overexpressed SLC6A17[G162R]-HA was still localized on SVs (*Figure 7—figure supplement 1A*, E). Overexpressed Syp-HA was previously reported to be localized on SVs (*Chantranupong et al., 2020*), which we confirmed (*Figure 7—figure supplement 1A*).

LC-MS analysis of contents of SVs immunoisolated by the anti-HA beads from OE-Syp-HA mice, when compared with immunoprecipitates from the brains of WT mice, would indicate SV enriched molecules. As expected, Glu, GABA, and ACh were all found to be enriched in these SVs (*Figure 7J, K, and L*), whereas Ser was not enriched (*Figure 7N*).

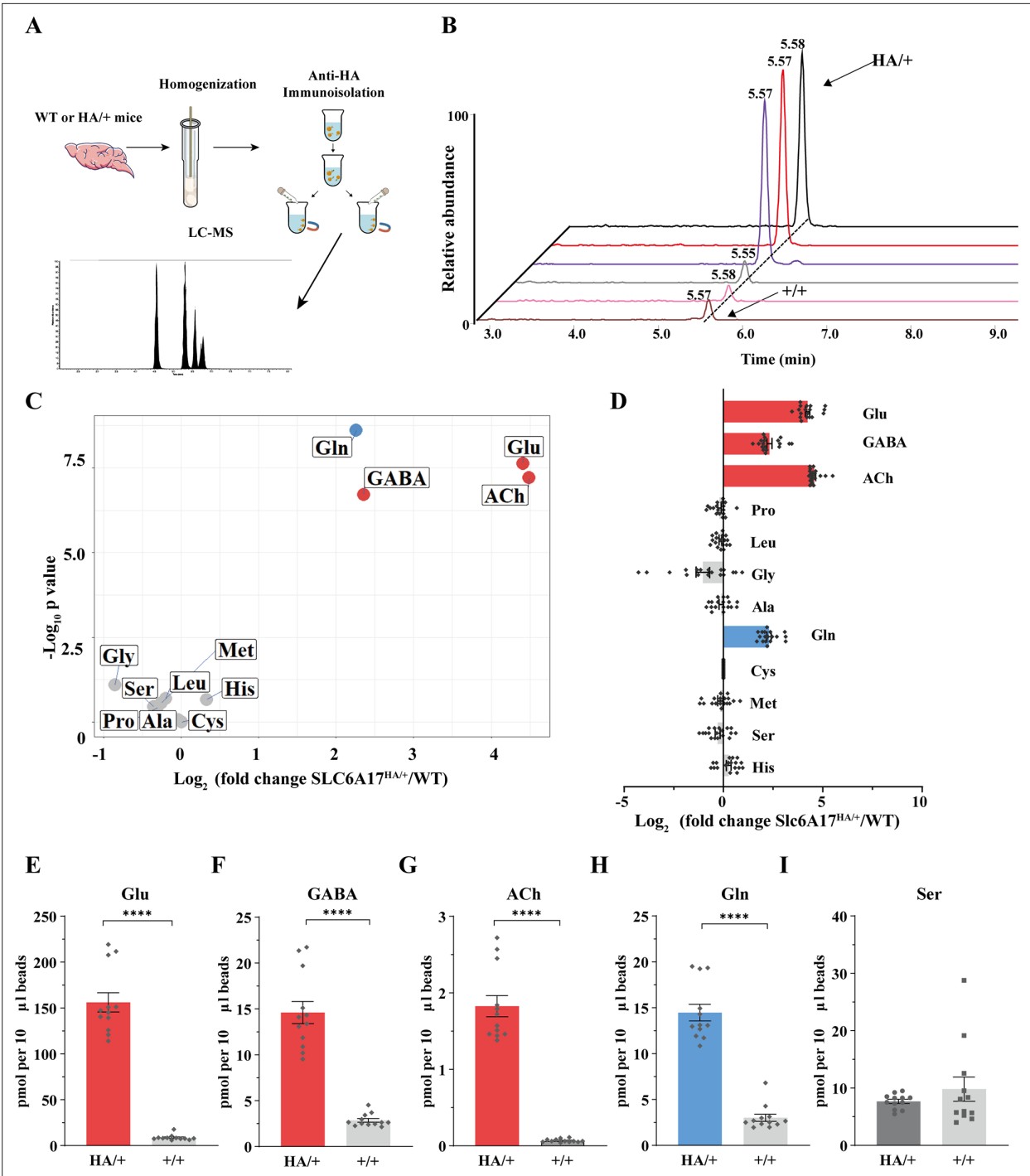

**Figure 6.** Glutamine (Gln) enrichment in synaptic vesicles (SVs) containing SLC6A17. (**A**) A schematic diagram illustrating the procedure to isolate SLC6A17-containing SVs for chemical analysis of SV contents. (**B**) Representative result showing MS Gln signals in SVs immunoisolated by the anti-HA beads in *Slc6a17*[HA/+] mice vs. those from *Slc6a17*[+/+] mice. (**C**) Volcano plot of chemical contents in the SVs immunoisolated by anti-HA beads from *Slc6a17*[HA/+] mice vs. *Slc6a17*[+/+] mice. The y axis shows p-values in $\log_{10}$ and the x axis shows the $\log_2$ of the ratio of the level of a molecule immunoisolated by anti-HA beads from *Slc6a17*[HA/+] mice vs. that from *Slc6a17*[+/+] mice. Classical neurotransmitters such as Glu, GABA, and ACh as well as the previously reported substrates of SLC6A17 are listed. (**D**) Ratios of the level of a chemical immunoisolated by anti-HA beads from *Slc6a17*[HA/+] mice vs. that from *Slc6a17*[+/+] mice (transformed into $\log_2$). (**E–I**) Contents of SLC6A17-containing SVs were quantified to mole per 10 μl anti-HA beads (n = 12, for each group from four different animals with three replicates each): Glu (**E**, p<0.0001 for *Slc6a17*[HA/+] vs. *Slc6a17*[+/+]); GABA (**F**, p<0.0001 for *Slc6a17*[HA/+] vs. *Slc6a17*[+/+]); ACh (**G**, p<0.0001 for *Slc6a17*[HA/+] vs. *Slc6a17*[+/+]); Gln (**H**, p<0.0001 for *Slc6a17*[HA/+] vs. *Slc6a17*[+/+]); Ser (**I**, p=0.7553 for *Slc6a17*[HA/+] vs. *Slc6a17*[+/+]).

*Figure 6 continued on next page*

*Figure 6 continued*

The online version of this article includes the following source data for figure 6:

**Source data 1.** Data points for *Figure 6C–I*.

The contents of SVs immunoisolated from OE-SLC6A17$^{G162R}$-HA mice, when compared with that from OE-Syp-HA and WT mice, would indicate molecules enriched in SVs containing SLC6A17$^{G162R}$ mutation, above the general contents of all SVs. Levels of Glu and GABA in SVs containing SLC6A17$^{G162R}$-HA were higher than those in WT mice (*Figure 7J and K*, *Figure 7—figure supplement 1B*), but not significantly different with those from OE-Syp-HA mice (*Figure 7J and K*). These results indicate that the subset of SVs containing SLC6A17$^{G162R}$-HA did not enrich Glu or GABA beyond the levels of Glu and GABA in the general population of SVs. The level of Gln in SVs containing SLC6A17$^{G162R}$-HA were much lower than that in from OE-SLC6A17-HA mice (*Figure 7M*), indicating that SLC6A17$^{G162R}$ was defective in transporting Gln into SVs in vivo. The level of Gln in SVs containing SLC6A17$^{G162R}$-HA was higher than that in WT mice, but not different from that in OE-Syp-HA mice (*Figure 7M*). These results indicate that Gln was present in SVs, but not enriched in SVs containing SLC6A17$^{G162R}$-HA comparing to the general populations of SVs expressing Syp-HA. The levels of the other eight AAs in SVs containing SLC6A17$^{G162R}$-HA were not significantly higher than those in WT mice (*Figure 7N*, *Figure 7—figure supplement 1B and D*).

We compared the contents of SVs immunoisolated from OE-SLC6A17-HA mice with those from either OE-Syp-HA mice or OE-SLC6A17$^{G162R}$-HA mice. The levels of GABA and ACh were not different among SVs from mice expressing SLC6A17-HA, Syp-HA, or SLC6A17$^{G162R}$-HA (*Figure 7I, I, K, and L*), indicating that SVs containing SLC6A17-HA were similar to the general populations of SVs. A moderate enrichment of Glu in SVs overexpressing SLC6A17-HA, compared to those overexpressing SLC6A17$^{G162R}$-HA, was observed (*Figure 7H and J*). Gln was increased most dramatically in SVs overexpressing SLC6A17-HA, compared to those overexpressing Syp-HA (*Figure 7I and M*) or SLC6A17$^{G162R}$-HA (*Figure 7H and M*). The levels of Ser (*Figure 7N*), Pro, Gly, Met, Leu, Ala, and Cys were not significantly increased in SVs overexpressing SLC6A17-HA, compared to those overexpressing Syp-HA (*Figure 7I*) or SLC6A17$^{G162R}$-HA (*Figure 7H*).

Taken together, our data provide in vivo evidence that the pathogenic mutation *SLC6A17*$^{G162R}$ could not transport Gln into SVs in the mouse brain, supporting the functional significance of Gln in the SVs.

## Requirement of SLC6A17 for Gln transport into the SV in *Slc6a17* knockout mice

We have found *Slc6a17*-KO mice to be defective in learning and memory (*Figure 2B–H*) and SLC6A17 protein was exclusively decreased on SVs. We then investigated Gln level in these mice. The anti-Syp antibody was used to immunoisolate SVs from the brains of *Slc6a17*$^{+/+}$, *Slc6a17*$^{+/-}$ and *Slc6a17*$^{-/-}$ mice (*Figure 8A–H*).

LC-MS analysis of SVs revealed that Gln was the only molecule found to be decreased in SVs from *Slc6a17*$^{-/-}$ mice, compared with *Slc6a17*$^{+/+}$ or *Slc6a17*$^{+/-}$ mice (*Figure 8H*). Quantitative analysis showed that the levels of classic transmitters such as Glu (*Figure 8A*), GABA (*Figure 8B*), and ACh (*Figure 8C*) were not significantly different among the SVs immunoisolated from the *Slc6a17*$^{+/+}$, the *Slc6a17*$^{+/-}$, or the *Slc6a17*$^{-/-}$ mice. The levels of the other eight AAs were also not significantly different among SVs immunoisolated from all three genotypes (*Figure 8D–F, G, I– L*). These results further support that Gln was transported into the SVs by SLC6A17 in vivo.

## Physiological requirement of SLC6A17 for Gln presence in the SVs in adult mice

To investigate whether the endogenous SLC6A17 is required in adulthood for Gln presence in the SVs, we used a viral-mediated CRISPR-based approach to remove the *Slc6a17* gene from adult mice.

Briefly, *Slc6a17*$^{-HA-2AiCre}$ mice were crossed with LSL-Cas9 transgene-carrying mice (*Platt et al., 2014*) and injected with an AAV-PHP.eB to express Syp-HA and guide RNAs (gRNA) targeting *Slc6a17* (AAV-PHP.eb-hSyn-DIO-Syp-HA-U6-sgRNAs) (*Figure 9A*, *Figure 9—figure supplement 1A*). A tandemly arrayed tRNA-gRNA structure was used to maximize the cleavage possibility in vivo (*Port and Bullock,*

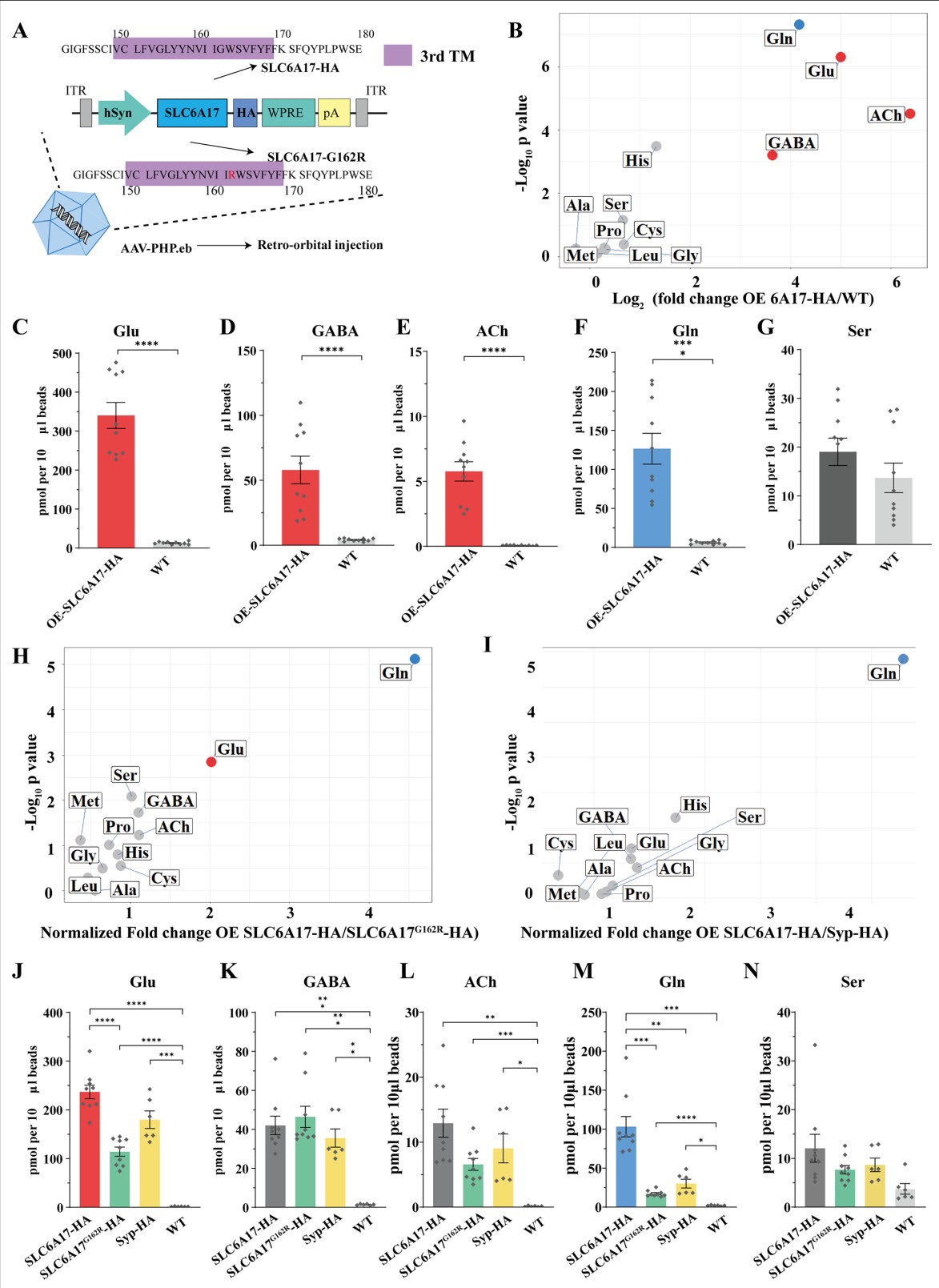

**Figure 7.** Increased glutamine (Gln) transport into synaptic vesicles (SVs) containing SLC6A17 but not into SVs containing SLC6A17[G162R]. (**A**) A schematic diagram illustrating the strategy for the AAV-PHP.eb virus-mediated in vivo overexpression of SLC6A17-HA and SLC6A17[G162R]-HA. (**B**) Volcano plot of chemical contents in the SVs isolated by anti-HA beads from the OE-SLC6A17-HA mice vs. WT mice. The y axis shows p-values in log10 and the x axis shows the log2 of the ratio of the level of a molecule immunoisolated by anti-HA beads from mice overexpressing SLC6A17-HA vs. that from WT mice.

*Figure 7 continued on next page*

*Figure 7 continued*

Classical neurotransmitters and previously reported substrates of SLC6A17 are listed. (**C–G**) Contents of OE-SLC6A17-containing SV are quantified to mole per 10 µl HA beads (n = 10, for each group from three different animals with three replicates in two animal and four replicates in one animal): Glu (**C**, p<0.0001 for OE-SLC6A17-HA vs. WT); GABA (**D**, p<0.0001 for OE-SLC6A17-HA vs. WT); ACh (**E**, p<0.0001 for OE-SLC6A17-HA vs. WT); Gln (**F**, p<0.0001 for OE-SLC6A17-HA vs. WT); (**G**, p=0.1655 for OE-SLC6A17-HA vs. WT). (**H**) Volcano plot comparing the chemical contents of SVs containing OE-SLC6A17-HA with those containing OE-SLC6A17$^{G162R}$-HA. Classical neurotransmitters and nine putative substrates of SLC6A17 are listed. Glu is the only classical transmitter significantly increased. Gln is the only substrates significantly increased in SLC6A17-HA containing SVs vs. SLC6A17$^{G162R}$-HA containing SVs. (**I**) Volcano plot comparing the chemical contents of SVs containing OE-SLC6A17-HA with those containing OE-Syp-HA. Classical neurotransmitters and nine putative substrates of SLC6A17 are listed. Gln is the only substrates significantly increased in SLC6A17 containing SVs vs. Syp-HA containing SVs. (**J–N**) Contents of SVs containing SCL6A17-HA, SVs containing OE-SLC6A17$^{G162R}$-HA, SVs containing Syp-HA immunosilated by anti-HA immunoisolation with that from WT mouse brains were quantified to mole per 10 µl HA beads and normalized to Syb2 relative abundance in WB (n = 9, 9, 6, and 6 for OE-SLC6A17-HA, OE-SLC6A17$^{G162R}$-HA, OE-Syp-HA, and WT, respectively, from different animals with three replicates each): Glu (**J**, p<0.0001 for OE-SLC6A17-HA vs. OE-SLC6A17$^{G162R}$-HA; p<0.0001 for OE-SLC6A17-HA vs. WT; p<0.0001 for OE-SLC6A17$^{G162R}$-HA vs. WT; p=0.0009 for OE-Syp-HA vs. WT; p=0.1551 for OE-SLC6A17-HA vs. OE-Syp-HA; p=0.0618 for OE-SLC6A17$^{G162R}$-HA vs. OE-Syp-HA); GABA (**K**, p=0.0002 for OE-SLC6A17-HA vs. WT; p=0.0002 for OE-SLC6A17$^{G162R}$-HA vs. WT; p=0.0039 for OE-Syp-HA vs. WT; p=0.9885 for OE-SLC6A17-HA vs. OE-SLC6A17$^{G162R}$-HA; p=0.9000 for OE-SLC6A17-HA vs. OE-Syp-HA; p=0.5840 for OE-SLC6A17$^{G162R}$-HA vs. OE-Syp-HA); ACh (**L**, p=0.0019 for OE-SLC6A17-HA vs. WT; p=0.0006 for OE-SLC6A17$^{G162R}$-HA vs. WT; p=0.0467 for OE-Syp-HA vs. WT; p=0.1054 for OE-SLC6A17-HA vs. OE-SLC6A17$^{G162R}$-HA; p=0.7563 for OE-SLC6A17-HA vs. OE-Syp-HA; p=0.8702 for OE-SLC6A17$^{G162R}$-HA vs. OE-Syp-HA); Gln (**M**, p=0.001 for OE-SLC6A17-HA vs. OE-SLC6A17$^{G162R}$-HA; p=0.0018 for OE-SLC6A17-HA vs. OE-Syp-HA; p=0.0003 for OE-SLC6A17-HA vs. WT; p<0.0001 for OE-SLC6A17$^{G162R}$-HA vs. WT; p=0.0189 for OE-Syp-HA vs. WT; p=0.2749 for OE-SLC6A17$^{G162R}$-HA vs. OE-Syp-HA); Ser (**N**, p=0.626 for OE-SLC6A17-HA vs. OE-SLC6A17$^{G162R}$-HA; p=0.8551 for OE-SLC6A17-HA vs. OE-Syp-HA; p=0.9874 for OE-SLC6A17-HA vs. OE-Syp-HA; p<0.0865 for OE-SLC6A17$^{G162R}$-HA vs. WT; p=0.1017 for OE-Syp-HA vs. WT).

The online version of this article includes the following source data and figure supplement(s) for figure 7:

**Source data 1.** Data points for *Figure 7B–I*.

**Figure supplement 1.** Virally mediated overexpression of SLC6A17-HA and SLC6A17$^{G162R}$-HA.

**Figure supplement 1—source data 1.** Original files of the full raw unedited blots for *Figure 7—figure supplement 1A and E*.

**Figure supplement 1—source data 2.** Uncropped blots with the relevant bands labeled for *Figure 7—figure supplement 1A and E*.

**Figure supplement 1—source data 3.** Data points for *Figure 7—figure supplement 1B–D*.

---

*2016*; *Xie et al., 2015*; *Figure 9—figure supplement 1A*). This strategy caused the excision of the *Slc6a17* gene and simultaneous expression of Syp-HA to tag SVs specifically in *Slc6a17*-expressing cells of adult mice (*Figure 9—figure supplement 1A*). *Slc6a17*-HA was significantly reduced both in the homogenates of total brains and in immunoisolated SVs from AAV-sgRNA/*Slc6a17*$^{iCre}$/Cas9$^+$ mice (*Figure 9B*, *Figure 9—figure supplement 1B*).

Importantly, when the SV contents from the brains of knockout (AAV-sgRNA/*Slc6a17*$^{iCre}$/Cas9$^+$) and control (AAV-sgRNA/*Slc6a17*$^{iCre}$/Cas9$^-$) mice were compared after LC-MS analysis, Gln was the only vesicular content significantly decreased in SVs from the knockout mice (*Figure 9C, D, and H*).

To our surprise, GABA and Glu were moderately increased (*Figure 9C–F*). The level of ACh, and those of the other eight AAs (Ala, Gly, Leu, Pro, Cys, Gly, His, and Ser), was not significantly changed in the SVs from adult *Slc6a17* knockout mice (*Figure 9C, D, G, and I*). The increased levels of Glu and GABA in SVs of virally mediated KO mice were different from the absence of Glu and GABA increases in the straightforward *Slc6a17* KO (*Figure 8A and B*), making the significance of the observed increases of Glu and GABA in the virally mediated *Slc6a17* knockout unclear.

These results indicate that SLC6A17 is physiologically necessary for Gln in SVs, but not for the other eight AAs.

## Functional significance of Gln in SVs supported by analysis of mice carrying a mutation mimicking a human ID mutation

To further address the question whether Gln transport into the SVs was functionally important, we examined the SVs from *Slc6a17*$^{P633R}$ mutant mice (*Figure 10A–L*). This mutation mimicked one of the human ID mutations (*Iqbal et al., 2015*), and its behavioral phenotypes have been found by us (*Figure 3*). We have shown that the SLC6A17$^{P633R}$ protein was not localized in the SVs (*Figure 3B*, *Figure 3—figure supplement 1D*).

LC-MS analysis of contents from SVs showed a significant decrease in Gln level in *Slc6a17*$^{P633R/P633R}$, compared to *Slc6a17*$^{+/+}$ and *Slc6a17*$^{P633R/+}$ mice (*Figure 10H*). The difference in Gln between *Slc6a17*$^{+/+}$ and *Slc6a17*$^{P633R/+}$ was not statistically significant (*Figure 10H*). The levels of Glu (*Figure 10A*),

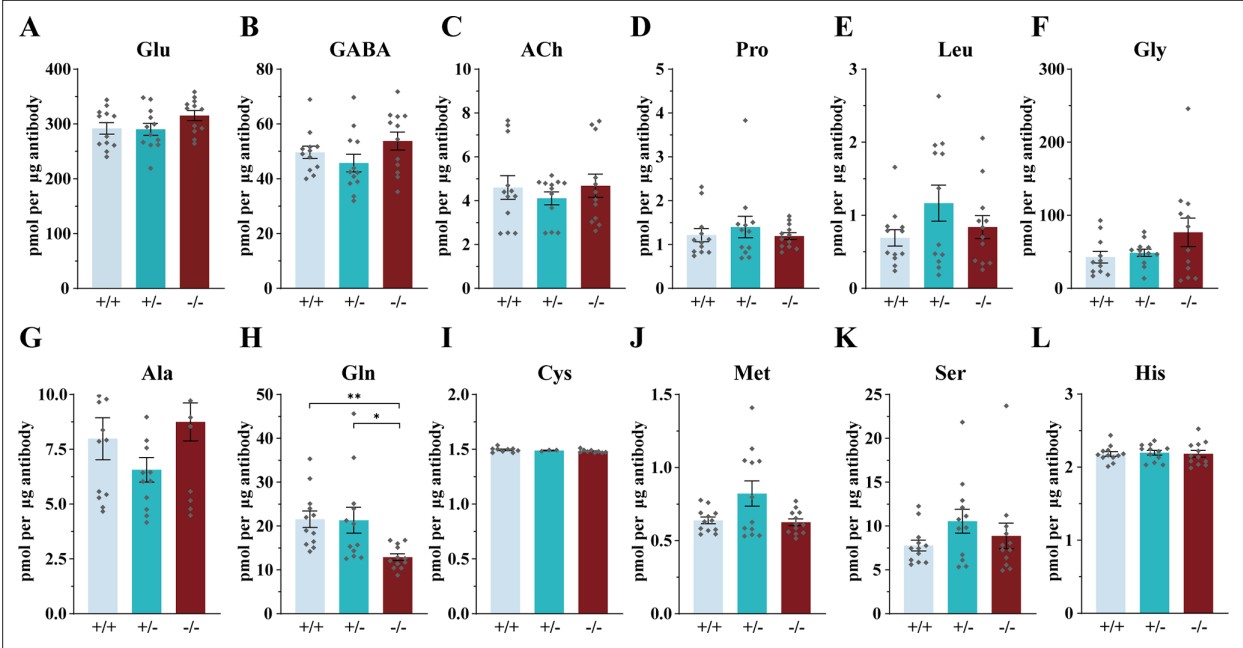

**Figure 8.** Reduced levels of glutamine (Gln) in the synaptic vesicles (SVs) of *Slc6a17* KO mice. Contents of SVs purified by the anti-Syp antibody from *Slc6a17*-KO mice were quantified to mole per 10 μg antibody (n = 12 for *Slc6a17*⁺/⁺, *Slc6a17*⁺/⁻, *Slc6a17*⁻/⁻, from four different animals with three replicates each): Glu (**A**, p=0.2749 for *Slc6a17*⁺/⁺ vs. *Slc6a17*⁻/⁻, p=0.2466 for *Slc6a17*⁺/⁻ vs. *Slc6a17*⁻/⁻); GABA (**B**, p=0.665 for *Slc6a17*⁺/⁺ vs. *Slc6a17*⁻/⁻, p=0.2503 for *Slc6a17*⁺/⁻ vs. *Slc6a17*⁻/⁻); ACh (**C**, p=0.9993 for *Slc6a17*⁺/⁺ vs. *Slc6a17*⁻/⁻, p=0.7186 for *Slc6a17*⁺/⁻ vs. *Slc6a17*⁻/⁻); Pro (**D**, p=0.9986 for *Slc6a17*⁺/⁺ vs. *Slc6a17*⁻/⁻, p=0.8035 for *Slc6a17*⁺/⁻ vs. *Slc6a17*⁻/⁻); Leu (**E**, p=0.826 for *Slc6a17*⁺/⁺ vs. *Slc6a17*⁻/⁻, p=0.61 for *Slc6a17*⁺/⁻ vs. *Slc6a17*⁻/⁻); Gly (**F**, p=0.3211 for *Slc6a17*⁺/⁺ vs. *Slc6a17*⁻/⁻, p=0.442 for *Slc6a17*⁺/⁻ vs. *Slc6a17*⁻/⁻); Ala (**G**, p=0.9092 for *Slc6a17*⁺/⁺ vs. *Slc6a17*⁻/⁻, p=0.1321 for *Slc6a17*⁺/⁻ vs. *Slc6a17*⁻/⁻); Gln (**H**, p=0.002 for *Slc6a17*⁺/⁺ vs. *Slc6a17*⁻/⁻, p=0.0489 for *Slc6a17*⁺/⁻ vs. *Slc6a17*⁻/⁻); Cys (**I**, p=0.3753 for *Slc6a17*⁺/⁺ vs. *Slc6a17*⁻/⁻, p=0.5718 for *Slc6a17*⁺/⁻ vs. *Slc6a17*⁻/⁻); Met (**J**, p=0.9735 for *Slc6a17*⁺/⁺ vs. *Slc6a17*⁻/⁻, p=0.13 for +/- vs. *Slc6a17*⁻/⁻); Ser (**K**, p=0.8678 for *Slc6a17*⁺/⁺ vs. *Slc6a17*⁻/⁻, p=0.7834 for *Slc6a17*⁺/⁻ vs. *Slc6a17*⁻/⁻); His (**L**, p>0.9999 for *Slc6a17*⁺/⁺ vs. *Slc6a17*⁻/⁻, p=0.9922 for *Slc6a17*⁺/⁻ vs. *Slc6a17*⁻/⁻).

The online version of this article includes the following source data for figure 8:

**Source data 1.** Data points for *Figure 8A–L*.

GABA (*Figure 10B*), and ACh (*Figure 10C*) in SVs were not significantly different among *Slc6a17*⁺/⁺, *Slc6a17*^P633R/+, and *Slc6a17*^P633R/P633R mice. The levels of the other eight AAs in the SVs were also not significantly different among *Slc6a17*⁺/⁺, *Slc6a17*^P633R/+, and *Slc6a17*^P633R/P633R mice (*Figure 10D–G, I–L*).

Thus, mislocalization of SLC6A17^P633R outside the SVs correlates with the inability of Gln enrichment in SVs (*Figure 10H*), as well as behavioral impairment (*Figure 3D, F, H, and I*), establishing the significance of SLC6A17 localization on the SVs.

## Discussion

We have taken multiple approaches to analyze seven kinds of genetically modified mice (four modified in the germline and one in adult brains) and our results indicating that Gln as an endogenous substrate of SLC6A17 and SLC6A17 is necessary and sufficient for Gln presence in SVs in vivo.

We have discovered the presence of Gln in the SVs and shown that it is functionally important because two mutations pathogenic for human ID patients both resulted in reduction of Gln in the SVs. We discuss these findings in basic neurobiology with implications in human ID pathology.

### The discovery of Gln in SVs and its functional significance

With behavioral studies in two types of *Slc6a17* mutant mice, we have shown that one ID pathogenic *Slc6a17* mutation (*Slc6a17*^P633R) causes defective learning and memory (*Figure 3*), similar to the phenotype of *Slc6a17* knockout mice (*Figure 2*). With biochemical purifications and genetically assisted EM, we have confirmed SLC6A17 localization on SVs (*Figure 4*).

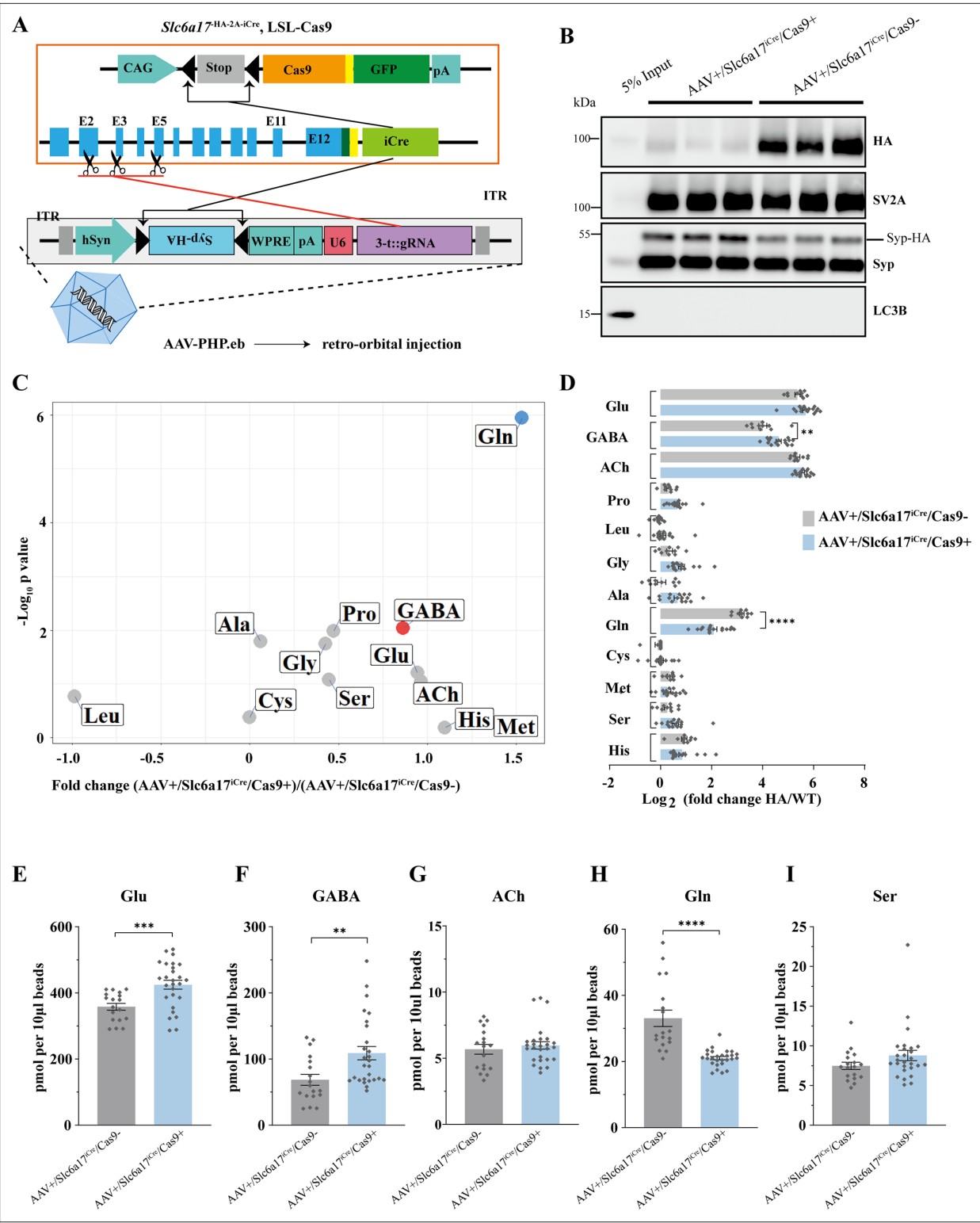

**Figure 9.** Physiological requirement of SLC6A17 for glutamine (Gln) transport into synaptic vesicles (SVs) in vivo. (**A**) A schematic diagram illustrating the strategy of Cas9-mediated cleavage of *Slc6a17* specifically in *Slc6a17*-positive neurons, and simultaneous labeling of all SVs in these neurons by Syp-HA. (**B**) Immunoblot results showing SLC6A17 protein was significantly reduced in targeted neurons, while Syp-HA was efficiently tagged onto the SVs of these neurons. (**C**) Volcano plot of contents of SVs from AAV-sgRNA/*Slc6a17*[iCre]/Cas9[+] targeted neurons compared to contents of SVs from control (AAV-sgRNA /*Slc6a17*[iCre]/Cas9[-]) neurons. Glu, GABA, ACh, and the nine previously reported substrates of SLC6A17 are listed. (**D**) Ratios of the level of a molecule from the SVs of AAV-sgRNA /*Slc6a17*[iCre]/Cas9[+] neurons vs. the level of the same molecule from SVs of AAV-sgRNA/*Slc6a17*[iCre]/Cas9[-] neurons

*Figure 9 continued on next page*

*Figure 9 continued*

shown as fold change (log$_2$ transformed). GABA level was significantly increased (p=0.0034 for AAV-sgRNA /*Slc6a17*$^{iCre}$/Cas9$^+$ vs. AAV-sgRNA /*Slc6a17*$^{iCre}$/Cas9$^-$). Gln level was significantly decreased (p<0.0001 for AAV-sgRNA /*Slc6a17*$^{iCre}$/Cas9$^+$ vs. AAV-sgRNA /*Slc6a17*$^{iCre}$/Cas9$^-$). (**E–H**) Contents of SVs from *Slc6a17* containing neurons were quantified to mole per 10 µl HA beads (n = 18, 27 for *Slc6a17*$^{iCre}$/Cas9$^-$ and *Slc6a17*$^{iCre}$/Cas9$^+$, respectively, from six and nine different animals with three replicates each): Glu (**E**, p=0.0005 for AAV-sgRNA /*Slc6a17*$^{iCre}$/Cas9$^+$ vs. AAV-sgRNA /*Slc6a17*$^{iCre}$/Cas9$^-$); GABA (**F**, p=0.0032 for AAV-sgRNA /*Slc6a17*$^{iCre}$/Cas9$^+$ vs. AAV-sgRNA /*Slc6a17*$^{iCre}$/Cas9$^-$); Gln (**G**, p<0.0001 for AAV-sgRNA /*Slc6a17*$^{iCre}$/Cas9$^+$ vs. AAV-sgRNA /*Slc6a17*$^{iCre}$/Cas9$^-$); Ser (**H**, p=0.0979 for AAV-sgRNA /*Slc6a17*$^{iCre}$/Cas9$^+$ vs. AAV-sgRNA /*Slc6a17*$^{iCre}$/Cas9$^-$).

The online version of this article includes the following source data and figure supplement(s) for figure 9:

**Source data 1.** Data points for *Figure 9C-I*.

**Source data 2.** Original files of the full raw unedited blots for *Figure 9B*.

**Source data 3.** Uncropped blots with the relevant bands labeled for *Figure 9B*.

**Figure supplement 1.** CRISPR/Cas9-mediated *Slc6a17* gene cleavage in *Slc6a17* expressing cells of adult mice.

**Figure supplement 1—source data 1.** Original files of the full raw unedited blots for *Figure 9—figure supplement 1B*.

**Figure supplement 1—source data 2.** Uncropped blots with the relevant bands labeled for *Figure 9—figure supplement 1B*.

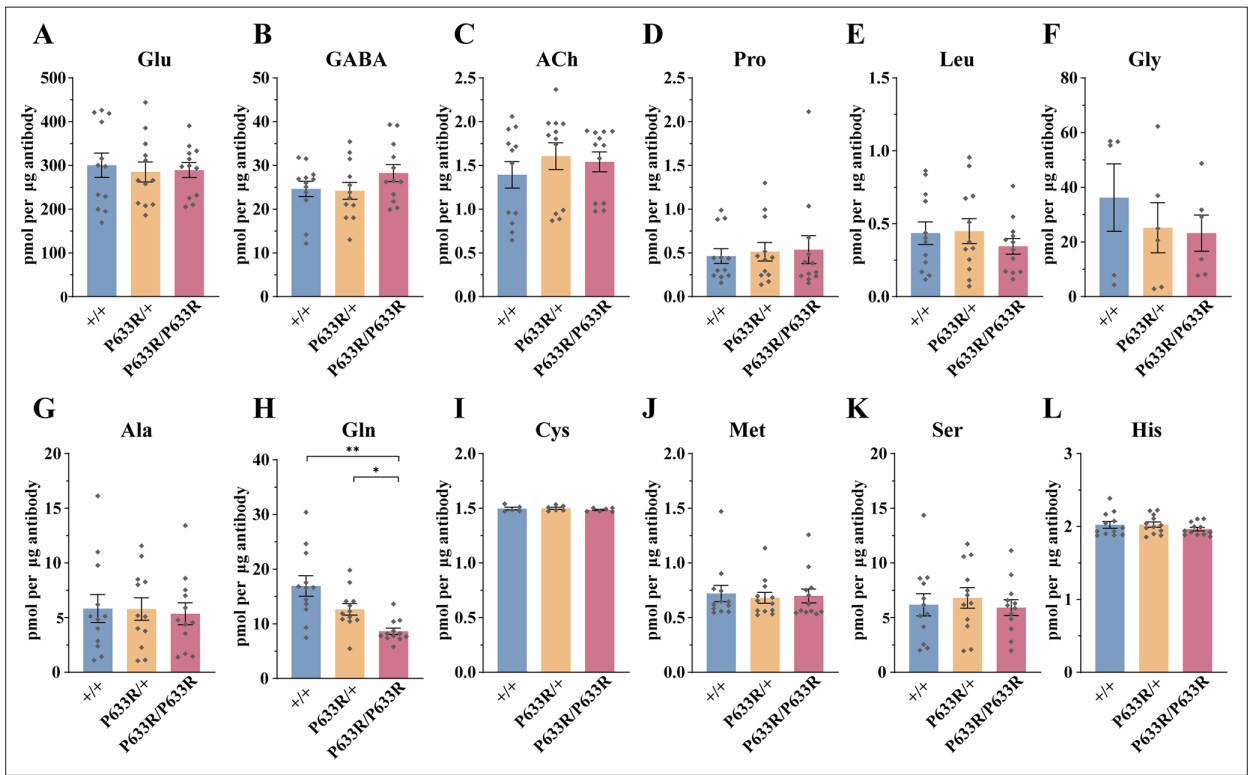

**Figure 10.** Reduced levels of glutamine (Gln) in the synaptic vesicles (SVs) of *Slc6a17*$^{P633R}$ mice. Contents of SVs purified by the anti-Syp antibody from *Slc6a17*$^{P633R}$ mice were quantified to mole per 10 µg antibody (n = 12 for all genotype, from four different animals with three replicates each): Glu (**A**, p=0.9804 for *Slc6a17*$^{+/+}$ vs. *Slc6a17*$^{P633R/P633R}$, p=0.9982 for *Slc6a17*$^{P633R/+}$ vs. *Slc6a17*$^{P633R/P633R}$); GABA (**B**, p=0.4432 for *Slc6a17*$^{+/+}$ vs. *Slc6a17*$^{P633R/P633R}$ p=0.3898 for *Slc6a17*$^{P633R/+}$ vs. *Slc6a17*$^{P633R/P633R}$); ACh (**C**, p=0.8175 for *Slc6a17*$^{+/+}$ vs. *Slc6a17*$^{P633R/P633R}$, p=0.9804 for *Slc6a17*$^{P633R/+}$ vs. *Slc6a17*$^{P633R/P633R}$); Pro (**D**, p=0.9674 for *Slc6a17*$^{+/+}$ vs. *Slc6a17*$^{P633R/P633R}$, p=0.9989 for *Slc6a17*$^{P633R/+}$ vs. *Slc6a17*$^{P633R/P633R}$); Leu (**E**, p=0.7116 for *Slc6a17*$^{+/+}$ vs. *Slc6a17*$^{P633R/P633R}$, p=0.6696 for *Slc6a17*$^{P633R/+}$ vs. *Slc6a17*$^{P633R/P633R}$); Gly (**F**, p=0.7376 for *Slc6a17*$^{+/+}$ vs. *Slc6a17*$^{P633R/P633R}$, p=0.9971 for *Slc6a17*$^{P633R/+}$ vs. *Slc6a17*$^{P633R/P633R}$); Ala (**G**, p=0.9871 for *Slc6a17*$^{+/+}$ vs. *Slc6a17*$^{P633R/P633R}$, p=0.986 for *Slc6a17*$^{P633R/+}$ vs. *Slc6a17*$^{P633R/P633R}$); Gln (**H**, p=0.0029 for *Slc6a17*$^{+/+}$ vs. *Slc6a17*$^{P633R/P633R}$, p=0.0117 for *Slc6a17*$^{P633R/+}$ vs. *Slc6a17*$^{P633R/P633R}$); Cys (**I**, p=0.8143 for *Slc6a17*$^{+/+}$ vs. *Slc6a17*$^{P633R/P633R}$, p=0.5475 for *Slc6a17*$^{P633R/+}$ vs. *Slc6a17*$^{P633R/P633R}$); Met (**J**, p=0.9942 for *Slc6a17*$^{+/+}$ vs. *Slc6a17*$^{P633R/P633R}$, p=0.995 for *Slc6a17*$^{P633R/+}$ vs. *Slc6a17*$^{P633R/P633R}$); Ser (**K**, p=0.9955 for *Slc6a17*$^{+/+}$ vs. *Slc6a17*$^{P633R/P633R}$, p=0.8355 for *Slc6a17*$^{P633R/+}$ vs. *Slc6a17*$^{P633R/P633R}$); His (**L**, p=0.6177 for *Slc6a17*$^{+/+}$ vs. *Slc6a17*$^{P633R/P633R}$, p=0.4316 for *Slc6a17*$^{P633R/+}$ vs. *Slc6a17*$^{P633R/P633R}$).

The online version of this article includes the following source data for figure 10:

**Source data 1.** Data points for *Figure 10A–L*.

We demonstrate the functional significance of the SV localization of SLC6A17 by showing that the human ID mutation *Slc6a17*[P633R] causes mislocalization of SLC6A17 outside the SVs (*Figure 3B*, *Figure 4—figure supplement 2A*). Gln transport into the SVs was decreased in *Slc6a17*[P633R] mutant mice (*Figure 10I*).

With multiple rounds of biochemical purification of SVs and chemical analysis of SV contents of genetically modified mice, we have found that Gln is present in SVs containing SLC6A17 and that SLC6A17 is sufficient for Gln presence in SVs (*Figures 5–7*). With two lines of germline mutant mice and virally mediated gene edited mice, we demonstrate that Gln is the only molecule reliably and reproducibly reduced in SVs when *Slc6a17* gene is deleted (*Figures 8 and 9*).

We have also characterized mice carrying a human ID mutation SLC6A17[G162R] and found it could not transport Gln into SVs (*Figure 7*), further supporting the importance of Gln in SVs. Thus, two mutations pathogenic in human ID are both LOF mutations: SLC6A17[P633R] with mislocalization outside the SVs and the SLC6A17[G162R] with defective Gln transport into the SVs.

The fact that all four kinds of *SLC6A17* LOF mutations (two KOs, two point mutations) have led to reduction in Gln in SVs strongly support the functional significance of Gln in SVs.

## Functional mechanisms of Gln in SVs

There are three possible functions for Gln in SVs: serving as a neurotransmitter, serving as a carbon source, and serving in a function so far unknown.

The first possibility is the most attractive. SLC6A17 is a member of the SLC6 family, which is also known as the NTT family because many members of this family are transporters for neurotransmitters (*Blakely and Edwards, 2012*; *Bröer, 2006*; *Kristensen et al., 2011*; *Rudnick et al., 2014*): SLC6A1 for GABA (*Guastella et al., 1992*; *Clark et al., 1992*; *Borden et al., 1994*; *Lopez-Corcuera et al., 1992*), SLC6A2 for noradrenaline (*Pacholczyk et al., 1991*), SLC6A3 for DA (*Giros et al., 1991*; *Kilty et al., 1991*; *Shimada et al., 1991*), SLC6A4 for 5-HT or serotonin (*Blakely et al., 1991*; *Hoffman et al., 1991*), and SLC6A5 and SLC6A9 for Gly (*Guastella et al., 1992*; *Smith et al., 1992*; *Liu et al., 1993*). Most of these are cytoplasmic transporters for transporting neurotransmitters from the synaptic cleft into the presynaptic cytoplasm.

The presence of the SLC6A17 protein on the vesicular membrane further supports the possibility that it is a neurotransmitter transporter because most of the vesicular transporters with known substrates transport neurotransmitters. If this is true for SLC6A17, then Gln would be a good candidate for a new neurotransmitter. However, this exciting possibility remains to be investigated. For Gln to be a transmitter, in addition to its presence in the SVs, more criteria should be satisfied: (1) it should be released from the nerve terminals upon electric stimulation, (2) it should be active on postsynaptic membrane, (3) it should be up-taken by a cytoplasmic transporter back into the presynaptic terminal or be removed otherwise (such as degraded enzymatically), and (4) a pharmacological blocker inhibiting its activity should interfere with chemical transmission in a physiologically significant context. Because Gln is abundant both intracellularly and extracellularly, we expect that demonstration of all the above would take some time, and likely by many laboratories.

There is a Glu/Gln cycle in which Glu released from neurons is taken up by glia and transformed into Gln, followed by Gln release from glia and taken up by neurons and transformed into Glu (*Bak et al., 2006*; *Benjamin and Quastel, 1972*; *Ottersen et al., 1992*; *van den Berg and Garfinkel, 1971*). Gln uptake across the neuronal cytoplasmic membrane relies on system A transporters (*Mackenzie et al., 2003*; *Reimer et al., 2000*; *Varoqui et al., 2000*; *Weiss et al., 2003*), while Gln uptake across the glia cytoplasmic membrane uses system L (*Deitmer et al., 2003*; *Kanai et al., 1998*), system N transporters (*Bröer and Brookes, 2001*; *Chaudhry et al., 2002*; *Chaudhry et al., 1999*), and system ASC (*Bröer et al., 2000*). It is unlikely that SLC6A17 and the Gln transports could be part of the Glu/Gln cycle because (1) the Glu/Gln cycle uses cytoplasmic transporters for Gln, not vesicular transporters for Gln; (2) Gln is transformed into Glu in the cytoplasm of neurons before being transported in the SVs as Glu; and (3) in all of our experiments presented here, when *SLC6A17* was genetically disrupted, Gln was consistently decreased, but Glu was either unchanged but not concomitantly increased, which was against the expectation of the Glu/Gln cycle. The only result in which Glu was increased when SLC6A17 was decreased came from an experiment with virally mediated genetic knockdown of SLC6A17 (*Figure 9E*), but GABA was also increased in that experiment (*Figure 9F*).

When the same experiment was carried with a germline genetic knockout of *SLC6A17*, Glu was not changed when SLC6A17 was decreased (*Figure 9A*).

The possibility of Gln as a carbon source is based on the fact that it is one of the major carbon sources (*Hensley et al., 2013*; *Reitzer et al., 1979*). It would be very curious if a carbon source should be stored in the SVs. It would suggest an entirely new function for Gln in the nerve terminal that has never been suspected before and may require considerable efforts before one can fully understand.

### Other substrates for SLC6A17

Of the nine AAs (Ala, Gly, Leu, Pro, Cys, Gln, Gly, His, and Ser) previously reported to be in vitro substrates for SLC6A17 transport in cell lines (*Parra et al., 2008*; *Zaia and Reimer, 2009*), we have reliably and reproducibly found evidence supporting only one (Gln) as present in vivo in all five types of experiments conducted with SV purifications: SVs immunoisolated from the WT brains by anti-Syp (*Figure 5*), SVs immunoisolated from the brains of mice with the HA tag fused in frame to the endogenous SLC6A17 protein (*Figure 6*), SVs immunoisolated from mice with virally introduced overexpression of Syp-HA, SVs immunoisolated from mice with virally introduced overexpression of SLC6A17$^{G162R}$-HA, and SVs immunoisolated from mice with virally introduced overexpression of SLC6A17-HA (*Figure 7*). While we did observe a moderate increase of His in one set of experiments (OE SLC6A17-HA vs. WT), we did not find evidence for its changes in other experiments.

Because multiple transporters can transport AAs, and the relative contribution of each transporter varies and the relative abundance of each small molecules and their detectability varies, we cannot rule out the possibility that SLC6A17 can contribute to the transport of other AAs into the SVs. However, our results can establish that Gln is transported in vivo.

In summary, integration of genetics, biochemistry, behavioral analysis, mass spectrometry, and electron microscopy has resulted in the discovery of Gln as a novel chemical in SVs, provided in vivo evidence that SLC6A17 is necessary and sufficient for Gln in SVs, generated animal models for, and suggested pathogenetic mechanism of, human ID. However, while we have answered the question how *Slc6a17* mutations may cause ID in animals, our research has also raised more questions. This natural process of questions begetting answers begetting questions is intrinsic to scientific inquiries and is a source of endless fun in conducting research.

## Materials and methods

### Reagents

Chemical reagents for biochemical purification and immunoisolation were all ordered form Sigma-Aldrich. The following antibodies were purchased from Synaptic Systemmouse anti-Synaptophysin 1 (Cat# 101011 for IP), rabbit anti-Synaptophysin 1 (Cat# 101002 for IB), rabbit anti-Synaptotagmin 1 (Cat# 105008 for IB), rabbit anti-VGLUT 1 (Cat# 135302 for IB), rabbit anti-VGAT (Cat# 131002 for IB), rabbit anti-VGLUT 2 (Cat# 135402 for IB), rabbit anti-Proton ATPase (Cat# 109002 for IB), rabbit anti-Synaptobrevin 2 (Cat# 104202 for IB), rabbit anti-SV2A (Cat# 119002 for IB), rabbit anti-SNAP23 (Cat# 111002 for IB), mouse anti-GluN1(Cat# 114011 for IB), and rabbit anti-ERC 1b/2 (Cat# 143003 for IB). The following antibodies were purchased from Abcam: mouse anti-Synaptophysin (Cat# ab52636 for IF), mouse Alexa Fluor 488 anti-Synaptophysin (Cat# ab196379 for IF), rabbit Anti-M6PR rabbit (Cat# ab124767 for IB), anti-Cathepsin D (Cat# ab75852 for IB), mouse anti-PSMC6 (Cat# ab22639 for IB), rabbit anti-GLUT4 (Cat# ab33780 for IB), and rabbit anti-Transferring receptor (Cat# ab84036 for IB). The following antibodies were purchased from Cell Signaling Technology: rabbit anti-LAMP2 (Cat# 49067 for IB), rabbit anti-EEA1 (Cat# 3288 for IB), rabbit anti-ERp72 (Cat# 5033 for IB), rabbit anti-Goglin-97 (Cat# 1292S for IB), rabbit anti-GM130 (Cat# 12480 for IB), rabbit anti-VDAC (Cat# 4661S for IB), rabbit anti-Syntaxin 6 (Cat# 2869 for IB), rabbit anti-LC3B (Cat# 2775 for IB), and rabbit anti-HA (Cat# 3724S for IF or IB). Mouse anti-PSD95 (Cat# 75028 for IB) were purchase from NeuroMab. Mouse anti-FLAG M2 HRP conjugated antibodies (Cat# A8592 for IB) were purchase from Sigma.

### Animals

All animal procedures were approved by the Animal Center of Peking University, and the experiments were carried out in accordance with the guidelines of Institutional Animal Care and Use Committee (IACUC) of Peking University. Mice were weaned at the age of 21 d. Mice were maintained under

standard conditions (12 hr light, 12 hr dark schedule). Room temperature was 23 ± 1°C. Humidity was 40–60%. Mice for behavior assays were 10–16 weeks old.

All transgenic mouse lines were generated in C57BL/6J background. Ai14 mice were a gift from Dr. Hongkui Zeng. Rosa26-LSL-Cas9 mice were obtained from The Jackson Laboratory (028551). *Slc6a17*-2A-CreERT2 and *Slc6a17*-KO mice were generated by CasGene biotech (Beijing, China). *Slc6a17*-HA-2A-iCre and *Slc6a17*P633R mice were generated by Biocytogen (Beijing, China). Mutant strains were generated by CRISPR/Cas9. Approximately 1.5 kb homologous arms upstream and downstream of target site were cloned into the targeting vector. The cleavage efficiency of single-guide RNAs (sgRNAs) targeting *SLC6A17* were tested in HEK cells. The sgRNAs selected for final injection were 5′-ccaacggacgctatggaagcggc-3′ for *Slc6a17*-2A-CreERT2 mice; 5′-catgcccaggtaacacctat ggg-3′ and 5′-cctgttaacaacactatctgaca-3′ for *Slc6a17*-KO mice; 5′-actgaggggtgctggccaagagg-3′ for *Slc6a17*-HA-2A-iCre mice; and 5′-ctgctagaatccaggaggcagg-3′ for *Slc6a17*P633R mice. All mutant strain were confirmed by PCR sequencing of the targeting regions. Southern blots were performed by Biocytogen for *Slc6a17*-HA-2A-iCre and *Slc6a17*P633R mice to rule out random insertions. For CreERT2 induction, tamoxifen was administrated intraperitoneally (i.p.) 1 mg/day/mice for five consecutive days at the age of 4–5 wk. Two weeks after tamoxifen injection, mice were killed for histology examination.

## Histology and immunocytochemistry

Mice were anesthetized, perfused, and post-fixed as described previously (*Liu et al., 2011*; *Qin and Luo, 2009*). Briefly, mice were deeply anesthetized with tribromoethanol (250 mg/kg; i.p.) and transcardially perfused with cold 0.1 M PBS followed by 4% paraformaldehyde (PFA) in 0.1 M PBS. Brains were fixed overnight in 4% PFA at 4°C and then immersed in 30% sucrose until it sank to the bottom at 4°C. It was embedded in the Tissue-Tek O.C.T. compound and frozen in a freezing microtome chamber (Leica CM 3050; Leica). Sections were 40 μm thick and collected on adhesion glass slides (P/N.80312–3161, CITOTEST). *Slc6a17*2A-CreERT2::Ai14 mice sections were washed with 0.3% Trtion X-100 in PBS (PBST) and then placed under cover-slips with a mounting solution (Fluoroshield with DAPI, F6057, Sigma). Sections were imaged by confocal microscope (Zesis LSM710) or Slide scanner (Zeiss Axio Scan. Z1). For immunofluorescence, freshly prepared slides were washed with PBST, followed by 1 hr 10% normal goat serum (in PBST) to block nonspecific binding. Primary antibodies were diluted to working concentrations in PBST and incubated with slides overnight at 4°C. Slides were washed with PBST and incubated with secondary antibodies for 1 hr at room temperature. After PBST washes, slides were placed under cover-slips with a mounting solution. For double staining in *Figure 3B* and *Figure 3—figure supplement 1C*, anti-HA antibody (RmAb, 1:200 dilution) was applied and labeled in Alexa Fluor 546, then Alexa Fluor 488 Anti-Syp (RmAb, ab196379, Abcam, 1:200 dilution) was applied. Slides were placed under cover-slips with a mounting solution and images were analyzed by confocal microscope (Zesis LSM710).

## Behavioral tests

Mice were placed in the experiment room for at least 1 hr before behavioral tests. All experiments were performed with the genotypes blind to the researchers. 70% ethanol was used to clear odor cues after each test.

### Open Field tests

Experiments were conducted in a square Plexiglas apparatus (40 × 40 × 35 cm) with illumination kept at 100–200 lux. Mice were gently placed in the center of the apparatus and allowed to explore for 30 min. Activities were recorded by a digital camera (HDR-XR550, SONY) set directly above the apparatus. After each trial, the apparatus was cleaned and the animals put back to home cage. Videos were analyzed by a home-made program created for *Slc6a17*-KO mice and Open Field (Mobile Datum, Shanghai, China) for *Slc6a17*P633R mice. The home-made program is based on OpenCV. Briefly, mice were extracted by subtracting background, which was updated for each frame in order to prevent environmental light shift. The center of the mouse was calculated, from which the distance was determined.

## Novel object recognition task

The experimental protocol was adopted from a previous report (*Bevins and Besheer, 2006*). On the first day, mice were individually habituated in an open box (40 × 40 × 25 cm). Next day, two identical objects (O) were placed into left and right corners of the box and mice were allowed to explore for 10 min. Object recognition was tested 1 hr later. One familiar object (O) was replaced by a novel object (N) before mice were placed back into the box. Mice were allowed to explore freely for 5 min. The time spent in the exploring the object was measured. Discrimination ratio was calculated by novel object interaction time/total interaction time with both objects. Testing sessions were recorded by a digital camera (HDR-XR550, Sony) and analyzed by software (Mobile Datum).

## Morris water maze

The Morris water maze task was modified according to previously described procedures (*Hitti and Siegelbaum, 2014*; *Qin et al., 2017*). The maze contained a round pool (diameter: 120 cm) and a round platform (diameter: 10 cm, 2 cm below the water surface). Water was 50 cm deep and set at 20 ± 1°C. Visual cues were placed above the pool. Titanium white was added into water to make it opaque for mice. We trained mice for four 1 min trials per day. If a mouse successfully found the platform, it was taken back to home cage 15 s after locating the platform. If a mouse failed to find the platform during 1 min, it was gently placed on platform for 15 s then taken back to the home cage. On days 1 and 2, cued learning was conducted to find the platform marked with a visible object. Mice were trained from different start points at each trial. On days 3–7, the object was removed while mice were trained with different start points to locate the hidden platform whose location was different with cue learning session. The latency for each animal to find the platform was recorded. On day 8, spatial memory was tested with a 1 min probe trial in which the platform was removed. Crossings into the platform area and total time spent in the target platform quadrant were recorded. The trajectories of probe trial were analyzed by Noldus Video Tracking Software (Ethovision XT 15, Noldus, Netherlands).

## Fear conditioning

A 3-day delay fear-conditioning protocol was employed to test fear memory (*Hitti and Siegelbaum, 2014*; *Qin et al., 2017*). A conditioning chamber (25 cm × 25 cm × 40 cm) with two types of contexts (context a: black walls, metal net floor for electrical stimulation; context b: white walls and white plastic floor) was used. On day 1, mice were trained in context a. Two rounds of training were performed. Each round consisted of a 1 min adaptation of environment, followed by a cued tone (30 s, 300 Hz, 90 dB sound) and a foot shock (2 s, 0.75 mA constant current). On day 2, contextual fear memory was assayed by placing the mice back in context a for 300 s. On day 3, mice were placed in context a. After 180 s, a training tone was sounded for 180 s. 5% acetic acid were used to clean chambers between day 3 and day 2. Two kinds of apparatus were used. *Slc6a17*-KO mice were tested in Fear Conditioning (Mobile Datum) and *Slc6a17*[P633R] mice were tested in Startle and Fear Combined System (Panlab, Spain). Animal activities were recorded and the percentage of time spent freezing (defined as the absence of all movement except for respiration) was measured by the software accordingly. Fear Conditioning (Mobile Datum) is a video-based recording system, so freezing time was reanalyzed by a double-blinded researcher to revise software variations. Startle and Fear Combined System (Panlab) is a gravity-based recording system.

## Elevated plus maze

The experimental protocol was adopted from previous reports (*Kouser et al., 2013*; *Powell et al., 2004*). Mice were placed in the center of a white plastic elevated plus maze, each arm was 33 cm long and 5 cm wide, with 15-cm-high opaque plastic walls on closed arms. Mice were allowed to explore for 5 min under the white light (~20 lux). Animal activities were recorded and analyzed by a software (Mobile Datum).

## **Subcellular fractionation**

Experimental protocols were based on previously described methods for SV purification (*Evans, 2015*; *Huttner et al., 1983*). Homogenate buffer (0.32 M sucrose and 5 mM HEPES-NaOH, pH 7.4, protease inhibitor cocktail [Complete EDTA-free, Roche]) was pre-cooled on ice, and all operations

were conducted on ice. Whole mouse brains were homogenized (kept as fraction H) in the homogenate buffer in a Potter Elvehjem homogenizer (Wheaton Instruments, USA, 9 strokes at 900 rpm) and centrifuged at 1000 × $g$ for 10 min. The supernatant was kept as Fraction S1 and pellet containing large membranes was resuspended (kept as Fraction P1). Fraction S1 was then centrifuged at 13,000 × $g$ for 20 min to yield a crude synaptosomal pellet (kept as Fraction P2), and remaining supernatant collected as Fraction S2. Synaptosomes in Fraction P2 were osmotic-lysed by adding 9 volumes of ice-cold ddH$_2$O (protease inhibitor cocktail added), followed by quick homogenization (3 strokes, 2000 rpm) and 1/20 volume 1 M HEPES-NaOH (pH 7.4) was added. The osmotic-lysed samples were incubated on ice for 40 min to maximize yields. Lysates were then centrifuged at 25,000 × $g$ for 20 min. Supernatant was collected as Fraction LS1 and pellet containing synaptic plasma membranes was resuspended (kept as Fraction LP1). Fraction LP2 was obtained after centrifugation of fraction LS1 for 2 hr at 165,000 × $g$ in a SW40 Ti rotor (Beckman, USA). Corresponding supernatant was collected as Fraction LS2. Further purification of the SVs in Fraction LP2 was performed by centrifugation on a continuous sucrose gradient. Fraction LP2 was resuspended in gradient buffer (40 mM sucrose, 5 mM HEPES-NaOH, pH 7.4) and 200 µl (1 mg/ml) resuspended fraction LP2 was layered onto a 11 ml 50–800 mM continuous sucrose gradient. The gradient was centrifuged at 65,000 × $g$ for 5 hr in SW40 Ti rotor and then divided to 11 equal fractions from the top to the bottom. 4 volumes of pre-cooled methanol (–20°C) was added into sub-fractions to precipitate proteins. Proteins were collected by centrifugation in 20,000 × $g$ for 20 min. All centrifugations were conducted at 4°C.

## Immunoisolation of SVs

Experimental protocols were adopted from previous established methods (**Burger et al., 1991**; **Burger et al., 1989**; **Martineau et al., 2013**). The entire experiments were conducted at 2–8°C. Whole brains were homogenized immediately after decapitation by ice-cold immunoisolation buffer (4 mM HEPES-KOH, 100 mM K$_2$-tartrate, 2 mM MgCl$_2$, pH 7.4, and protease inhibitor cocktail) in Potter Elvehjem homogenizer (15 strokes, 2000 rpm). Homogenates were centrifuged 35,000 × g for 25 min at 4°C. The supernatant (protein concentration 3 mg/ml) was used as input to incubated with magnetic beads coupled with antibody. For anti-Syp immunoisolation, anti-Syp (101 011, SySy) antibody was pre-incubated with Pierce protein G magnetic beads (88848, Thermo Scientific) in 0.1% BSA PBS at 4°C overnight. The anti-Syp antibody and mice IgG (02-6502, Invitrogen) were bound to Pierce protein G magnetic beads at the ratio of 33 µg of IgG per mg beads. For anti-HA immunoisolation, Pierce anti-HA magnetic beads (88837, Thermo Scientific) were used and washed just before each experiment. Beads were incubated with input supernatant for 2 hr with constant rotation at 4°C. Beads were washed six times by immunoisolation buffer after incubation. Following the final wash, beads were magnetically separated and vesicular contents were extracted by 50 µl 80%/20% methanol/H$_2$O (–80°C) with 5 µM internal standards. To maximize the extraction efficiency and protein precipitation rate, the extraction mix was transferred to –20°C for 1 hr. The extraction mix was then centrifuged for 20 min at 20,000 × $g$, and supernatant was transferred to new vials subjected to LC-MS. The remaining beads were extracted by SDS loading buffer for immunoblot analysis.

## Immunoblotting

Proteins were extracted from each sample with SDS loading buffer (3% SDS, 15% glycerol, 180 mM Tris-HCl, 4% β-mercaptoethanol). For subcellular fractionation samples, protein concentrations were determined by the BCA protein assay (Pierce, 23250, Thermo Scientific), and equal quantities of proteins were subjected to SDS-PAGE. For immunoisolation samples, equal volumes of SDS loading buffer were added to magnetic isolated beads. Samples were incubated in 75°C for 20 min to better immunoblot transmembrane proteins. Briefly, samples were loaded to 10% SDS-PAGE gels (TGX Fast-Cast Acrylamide Kits, 1610183, Bio-Rad) and transferred to 0.45 mm NC membranes (HATF00010, Millipore) for 2 hr. Membranes were subjected to 5% milk in TBST buffer (Tri-buffered Saline with Tween 20) at room temperature for 1 hr to block non-specific binding. Primary antibodies were prepared in TBST and incubated with membranes at 4°C overnight. Secondary antibodies were either anti-rabbit IgG HRP (A6154, Sigma, 1:5000 dilution) or anti-mouse IgG HRP (715-035-150, Jackson ImmunoResearch, 1:5000 dilution). Final results were visualized using chemiluminescence (Merck Millipore) in e-BLOT Touch Imager (e-BLOT Life Science, Shanghai).

## AAV plasmid construction and virus injection

AAV-PHP.eb viruses were packaged in HEK293T cells by the vector center of CIBR. The titers of AAVs were determined by real-time PCR. Viruses were subpackaged and stored in –80°C before use. All AAV constructs used for viral packaging were linearized by restriction enzymes to insert target sequences. Bacterial cultures were grown at 33°C. AAV-PHP.eb viruses were packaged by Vector Center in CIBR. Viruses were delivered retro-orbitally into the venous sinus of 6-week-old mice. For APEX2 staining, 2 * 10[11] particles of virus were injected. For gene overexpression and sgRNA expression, 1 * 10[11] particles of virus were injected. All plasmids will provided upon request. Sequence information is provided in *Supplementary file 1*.

## Construction of pAAV-hSyn-SLC6A17-APEX2-WPRE-pA plasmid

This plasmid consisted of human *SLC6A17* cDNA, 3×HA tag, V5 tag, and APEX2. *SLC6A17* cDNA was cloned from commercially available cDNA library (Vigene, Shandong, China). The 3×HA tag ( YPYDVPDYAGYPYDVPDYAGSYPYDVPDYA) were chosen the same version as *Slc6a17*[-HA-2A-iCre] mice was in-frame fused with cDNA by GS4 linker (GGGGSGGGGS GGGGSGGGGS). V5-APEX2 sequence were cloned from mito-V5-APEX2 plasmid (#72480, Addgene). V5-APRX2 were first cloned into pAAV-hSyn-WPRE-pA plasmid. Then *SLC6A17*-HA was inserted before V5-APEX2. NEBuilder HiFi DNA Assembly Cloning Kit (NEB, E5520S) were used to insert fragments to backbone plasmids.

## Construction of pAAV-hSyn-SLC6A17-HA-WPRE-pA plasmid and pAAV-hSyn-SLC6A17[G162R]-HA-WPRE-pA plasmid

*SLC6A17*-HA sequence consists of pAAV-hSyn-SLC6A17-APEX2. G162R point mutation was introduced by PCR (forward primer: 5'-attataatgtgatcatcAggtggagcatcttc-3'; reverse primer: 5'-Tgatgatc acattataatacagcccc-3') in a middle plasmid without ITR element. Then the *SLC6A17*[G163R]-HA sequence was cloned into pAAV-hSyn-WPRE-pA plasmid.

## Construction of pAAV-hSyn-DIO-Syp-HA-WPRE-pA-U6-3-t::gRNA WPRE-pA plasmid

tRNA-sgRNA design was modified from *Port and Bullock, 2016*; *Xie et al., 2015*. sgRNAs were chosen based on published database (*Michlits et al., 2020*). The sequence of selected gRNA: sgRNA1 (5'- CGATGCTCCAGGCCACAAGGAGG-3', targeting exon 5); sgRNA2 (5'-GCCGTGGCAGCATTGG TGTGTGG-3', targeting exon 3); sgRNA3 (5'-TGGGCCTGGGCAACATCTGGAGG-3', targeting exon 2). Syp-HA sequence was synthesized and cloned into AAV backbone with hSyn promoter and DIO cassette.

## Electron microscopy with APEX2

Procedures for APEX2 based EM labeling were modified from previous reports (*Lam et al., 2015*; *Martell et al., 2017*). Briefly, the AAV-injected animals were anesthetized and perfused with PBS and 4% PFA and 1% glutaraldehyde (GA) for pre-fixation. Mouse brains were dissected and cut into approximately 1 mm³ blocks and was added into 2.5% GA for post-fixation for 2 hr. Samples were then washed with 0.1 M phosphate buffer (PB). Following pre-incubation in 0.5 mg/ml 3,3-diaminobenzidine (DAB, D5905, Sigma) for 30 min, 0.03% $H_2O_2$ (final concentration) was added into samples for APEX2 oxidation (note that different sources of DAB had considerable disparity, and D5905 produced the best signal than other DABs in our lab), although the recommended D8001 was poorer than D5905 (*Ludwig, 2020*; *Martell et al., 2017*). The reaction was monitored by the dark brown color in tissue block and was stopped by removing the DAB solution and washed three times with 0.1 M PB. Most of the successful DAB staining usually turned deep color within 10 min. After DAB staining, samples were incubated in 4% osmium tetroxide in 0.1 M PB on ice for 1 hr in dark environment. Samples were washed with ddH$_2$O and incubated in 1% aqueous uranyl acetate overnight at 4°C. Next day, samples were dehydrated in graded ethanol series (30%, 50%, 75%, 85%, 95%, and 100%) at 4°C and rinsed once in acetone with each rinse for 10 min at room temperature. Samples were infiltrated in Epon 812 resin using a 1:3, 1:1, and 3:1 ratio of resin and acetone for 2 hr each followed by three infiltrations with 100% resin for 30 min each. Finally, samples were embedded in fresh resin in a vacuum oven for 24 hr at 65°C. Blocks were sectioned by UC7 ultramicrotome (Leica Microsystems, Vienna, Austria) into 70 nm. A 120 kV JEOL JEM-1400Flash TEM (JEOL, Japan) was used to image the brain

and images were acquired by an XAROSA camera (Emsis GmbH, Muenster, Germany). Images were saved in a .tif format and were further analyzed using FIJI-ImageJ 2.1.0/1.53c (Wayne Rasband, NIH). The segmentation of SVs was performed using the TrakEM2 v1.0a (Univ/ETH Zurich) plugin in FIJI and the mean intensity of DAB+ or SV was quantified using a custom-made macro. Figures were created using OriginPro 2022.

## LC-MS for chemical detection

LC-MS was used to detect and quantify the chemical contents of samples based on previous reports (*Abu-Remaileh et al., 2017*; *Contrepois et al., 2015*; *Mackay et al., 2015*; *Wernisch and Pennathur, 2016*; *Zhang et al., 2014*; *Zhang et al., 2012*). A Vanquish UHPLC system coupled to a Q Exactive HF-X mass spectrometer (both instrument Thermo Fisher Scientific, USA) were used for LC-MS analysis along with SeQuant ZIC-HILIC column (150 mm × 2.1 mm, 3.5 μm, Merck Millipore, 150442) in the positive mode and SeQuant ZIC-pHILIC column (150 mm × 2.1 mm, 5 μm, Merck Millipore, 150460) in the negative mode. For ZIC-HILIC column, the mobile phase A was 0.1% formic acid in water and the mobile phase B was 0.1% formic acid in acetonitrile. The linear gradient was as following: 0 min, 80% B; 6 min, 50% B; 13 min, 50% B; 14 min, 20% B; 18 min, 20% B; 18.5 min, 80% B; 30 min, 80% B. The flow rate used was 300 μl/min and the column temperature was maintained at 30°C. For ZIC-pHILIC column, the mobile phase A is 20 mM ammonium carbonate in water, adjusted to pH 9.0 with 0.1% ammonium hydroxide solution (25%) and the mobile phase B is 100% acetonitrile. The linear gradient was as follows: 0 min, 80% B; 2 min, 80% B; 19 min, 20% B; 20 min, 80% B; 30 min, 80% B. The flow rate used is 150 μl/min and the column temperature is 25°C. Samples were maintained at 4°C in Vanquish autosampler. 3 μl of extracted metabolites were injected for each run. IP samples were subjected to ZIC-HILIC column in positive mode for major metabolites detection, and then subject to ZIC-pHILIC column in negative mode for orthogonal detection.

Calibration was performed prior to analysis using the Pierce Positive and Negative Ion Calibration Solutions (Thermo Fisher Scientific) every 7 d. Acetonitrile was LC-MS grade (A955, Thermo Fisher Scientific), and all other regents were LC-MS Optima grade from sigma. Isotope-labeled amino acids (sigma) as internal extraction standards were supplemented into extraction solution. Orbitrap mass spectrometer was operated with the following parameters: scanning in Full MS mode (2 μscans) from 70 to 1050 m/z at 70,000 resolution, with 4 kV spray voltage (3.5 kV for negative mode), 50 shealth gas (35 for negative mode), 12 auxiliary gas (8 for negative mode), capillary temperature 300°C, S-lens RF level 55, AGC target 1E6, and maximum injection time 200 ms.

## TMT-labeled SV proteomics

Procedures for quantitative SV proteomics were modified from previous reports (*Boyken et al., 2013*; *Grønborg et al., 2010*). Proteins were labeled using TMT10plex Isobaric Label Reagent Set (90113, Thermo Scientific). In short, LP2 fractions were purified form both *Slc6a17*-KO +/+ and +/- mice, with the final step centrifuged in 230,000 × *g* instead of 165,000 × *g*. Purified LP2 were stored in –80°C for long-term storage. SVs were further purified by immunoisolation with the anti-Syp antibody coupled to magnetic protein G beads incubated with resuspended LP2 fraction for 2 hr at 4°C (LP2 should be repeatedly passed through 28 gauge needles to be fully resuspended). After six washes in PBS, SVs were magnetically separated and dissociated form beads by adding lysis buffer [10% SDS in 100 mM triethyl ammonium bicarbonate (TEAB)] to samples. Beads were heated in 70°C for 10 min to fully dissolve proteins and then centrifuged 10 min at 16,000 × *g* to remove beads. Protein concentrations were determined by BCA kits (23227, Thermo Scientific) to ensure the total proteins between 50 and 100 μg. Equal amounts of proteins were transferred into new tubes and adjusted to 100 μl with 100 mM TEAB. 5 μl 200 mM TCEP (3,3',3''-phosphinetriyltripropanoic acid) was added and incubated in 55°C for 1 hr. Then 5 μl freshly prepare 375 mM iodoacetamide were added to the samples and incubated for 30 min in dark environment at room temperature. Following addition of six volumes (~600 μl) of pre-chilled (–20°C) acetone, proteins were precipitated overnight at –20°C. Next day, samples were centrifuged at 8000 × *g* for 10 min at 4°C to precipitate proteins. After acetone removal, 100 μl of 100 mM TEAB were added to resuspend proteins. Trypsin was added at 2.5 μg per sample and incubated at 37°C. Then freshly made TMT Label Reagents (0.8 mg/41 μl anhydrous acetonitrile) were added to samples and incubated for 1 hr at room temperature. 8 μl 5% hydroxylamine were added to the samples and incubated for 15 min to quench the reaction. Equal amounts of

each sample were combined in a new centrifuge tube and labeled peptides dried in SpeedVac. Pierce High pH Reversed-Phase Peptide Fractionation Kit (84868, Thermo Scientific) were used to clean up and fractionate TMT-labeled peptides before LC-MS/MS. Peptides were identified and analyzed by the Proteomic Center of PKU.

## Statistical analyses

For statistical comparisons of two groups, we used the nonparametric Mann–Whitney test to compare two columns of data. For comparisons of more than two groups, we used the one-way ANOVA Brown–Forsythe ANOVA test, and Dunnett's T3 multiple-comparisons test. We used two-way ANOVA with Tukey's multiple-comparisons test for multiple factor comparison. An α level of 0.05 (two-tailed) was set for significance. All statistical analysis was done using Prism 8 (GraphPad Software).

## Acknowledgements

We are grateful to CIBR, Peking-Tsinghua Center for Life Sciences, Shenzhen Bay Laboratory, CAMS Innovation Fund for Medical Sciences (CIFMS) (2019-I2M-5-015), Changping Laboratory for support. We thank the Proteomics Core, EM Core in National Center for Protein Sciences at Peking University for technical support.

## Additional information

### Funding

| Funder | Grant reference number | Author |
|---|---|---|
| Chinese Institute for Brain Research | | Yi Rao |
| Peking-Tsinghua Center for Life Science | | Yi Rao |
| Shenzhen Bay Laboratory | | Yi Rao |
| Chinese Academy of Medical Sciences Initiative for Innovative Medicine | 2019-I2M-5-015 | Yi Rao |
| Changping laboratory | | Yi Rao |

The funders had no role in study design, data collection and interpretation, or the decision to submit the work for publication.

### Author contributions

Xiaobo Jia, Conceptualization, Resources, Data curation, Formal analysis, Investigation, Visualization, Methodology, Writing - original draft, Project administration, Writing – review and editing; Jiemin Zhu, Xiling Bian, Validation, Investigation, Methodology; Sulin Liu, Resources, Formal analysis, Investigation; Sihan Yu, Validation, Investigation; Wenjun Liang, Lifen Jiang, Investigation; Renbo Mao, Methodology; Wenxia Zhang, Resources, Supervision, Project administration; Yi Rao, Conceptualization, Supervision, Funding acquisition, Writing – review and editing

### Author ORCIDs

Xiaobo Jia  http://orcid.org/0000-0002-4214-8906
Sulin Liu  http://orcid.org/0000-0002-6776-2216
Yi Rao  http://orcid.org/0000-0002-0405-5426

### Ethics

All animal procedures were approved by the Animal Center of Peking University and Animal Care and Use Committee of the Chinese Institute for Brain Research (CIBR). Experiments were carried out in accordance with the guidelines of Institutional Animal Care and Use Committee (IACUC) of Peking University (LSC-RaoY-8) and CIBR (CIBR-IACUC-006).

Joint Public Review https://doi.org/10.7554/eLife.86972.2.sa1
Author Response https://doi.org/10.7554/eLife.86972.2.sa2

## Additional files

### Supplementary files

• Supplementary file 1. Sequence information file for AAV plasmid used in this article. pAAV-hSyn-SLC6A17-APEX2-WPRE-pA, pAAV-hSyn-SLC6A17-HA-WPRE-pA, pAAV-hSyn-SLC6A17G162R-HA-WPRE-pA, and pAAV-hSyn-DIO-Syp-HA-WPRE-pA-U6-3-t::gRNA WPRE-pA. Sequence information file for pAAV-hSyn-SLC6A17-APEX2-WPRE-pA.

• Supplementary file 2. Summarize table for Slc6a17 gene expression in *Slc6a17*[-2A-CreERT2] mice. Sequence information file for pAAV-hSyn-SLC6A17-HA-WPRE-pA.

• MDAR checklist

### Data availability

All data generated or analysed during this study are included in the manuscript and supporting files; Source Data files have been provided for Figures 2, 3, 4, 5, 6, 7, 8, 9, 10, Figure 2-figure supplement 1, Figure 2-figure supplement 2, Figure 3-figure supplement 1, Figure 3-figure supplement 2, Figure 4-figure supplement 1, Figure 5-figure supplement 1, Figure 7-figure supplement 1, and Figure 9-figure supplement 1.

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

# Appendix 1

**Appendix 1—key resources table**

| Reagent type (species) or resource | Designation | Source or reference | Identifiers | Additional information |
|---|---|---|---|---|
| Gene (*Homo sapiens*) | *SLC6A17* | GenBank | NM_001010898.4 | |
| Gene (*H. sapiens*) | Syp | GenBank | NM_009305.2 | |
| Strain, strain background (C57BJ/N) | Ai14 | Dr. Hongkui Zeng lab | | |
| Strain, strain background (C57BJ/N) | Rosa26-LSL-Cas9 | Jackson Laboratory | 028551 | |
| Strain, strain background (C57BJ/N) | *Slc6a17*-2A-CreERT2 | CasGene Biotech (Beijing, China) | | |
| Strain, strain background (C57BJ/N) | *Slc6a17*-KO | CasGene Biotech (Beijing, China) | | |
| Strain, strain background (C57BJ/N) | *Slc6a17*-HA-2A-iCre | Biocytogen (Beijing, China) | | |
| Strain, strain background (C57BJ/N) | *Slc6a17*P633R | Biocytogen (Beijing, China) | | |
| Recombinant DNA reagent | pAAV-hSyn-SLC6A17-APEX2-WPRE-pA (plasmids) | This paper | | *Supplementary file 1* |
| Recombinant DNA reagent | pAAV-hSyn-SLC6A17-HA-WPRE-pA (plasmids) | This paper | | |
| Recombinant DNA reagent | pAAV-hSyn-SLC6A17G 162R-HA-WPRE-pA (plasmids) | This paper | | |
| Recombinant DNA reagent | pAAV-hSyn-DIO-Syp-HA-WPRE-pA-U6-3-t::gRNA WPRE-pA (plasmids) | This paper | | |
| Recombinant DNA reagent | NEBuilder HiFi DNA Assembly Cloning Kit | NEB | E5520S | |
| Antibody | Anti-Synaptophysin (mouse monoclonal) | Synaptic System | 101011 | IP 1:200 |
| Antibody | Anti-Synaptophysin (rabbit polyclonal) | Synaptic System | 101 002 | IB 1:2000 |
| Antibody | Anti-Synaptotagmin 1 (rabbit polyclonal) | Synaptic System | 105 008 | IB 1:2000 |
| Antibody | Anti-VGLUT 1 (rabbit polyclonal) | Synaptic System | 135 302 | IB 1:2000 |
| Antibody | Anti-Proton ATPase (rabbit polyclonal) | Synaptic System | 109 002 | IB 1:2000 |
| Antibody | Anti-Synaptobrevin 2 (rabbit polyclonal) | Synaptic System | 104 202 | IB 1:2000 |
| Antibody | Anti-SV2A (rabbit polyclonal) | Synaptic System | 119 002 | IB 1:5000 |
| Antibody | Anti-VGAT (rabbit polyclonal) | Synaptic System | 131 002 | IB 1:2000 |
| Antibody | Anti-VGLUT2 (rabbit polyclonal) | Synaptic System | 135 402 | IB 1:2000 |
| Antibody | Anti-SNAP23 (rabbit polyclonal) | Synaptic System | 111 202 | IB 1:2000 |
| Antibody | Anti-GluN1 (mouse polyclonal) | Synaptic System | 114 011 | IB 1:5000 |
| Antibody | Anti-ERC 1b/2 (rabbit polyclonal) | Synaptic System | 143003 | IB 1:5000 |
| Antibody | Anti-Synaptophysin (mouse monoclonal) | Abcam | ab52636 | IF 1:500 |
| Antibody | Alexa Fluor 488 Anti-Synaptophysin (mouse monoclonal) | Abcam | ab196379 | IF 1:200 |

*Appendix 1 Continued on next page*

*Appendix 1 Continued*

| Reagent type (species) or resource | Designation | Source or reference | Identifiers | Additional information |
|---|---|---|---|---|
| Antibody | Anti-Cathepsin D (rabbit monoclonal) | Abcam | ab75852 | IB 1:2000 |
| Antibody | Anti-PSMC6 (mouse monoclonal) | Abcam | ab22639 | IB 1:5000 |
| Antibody | Anti-GLUT4 (rabbit monoclonal) | Abcam | ab33780 | IB 1:2000 |
| Antibody | Anti-Transferring receptor (rabbit monoclonal) | Abcam | ab84036 | IB 1:2000 |
| Antibody | Anti-M6PR (rabbit monoclonal) | Abcam | ab124767 | IB 1:2000 |
| Antibody | Anti-LAMP2 (rabbit monoclonal) | CST | #49067 | IB 1:2000 |
| Antibody | Anti-EEA1 (rabbit monoclonal) | CST | #3288 | IB 1:2000 |
| Antibody | Anti-ERp72 (rabbit monoclonal) | CST | #5033 | IB 1:2000 |
| Antibody | Anti-Goglin-97 (rabbit monoclonal) | CST | #12192S | IB 1:1000 |
| Antibody | Anti-GM130 (rabbit monoclonal) | CST | #12480 | IB 1:2000 |
| Antibody | Anti-VDAC (rabbit monoclonal) | CST | #4661S | IB 1:2000 |
| Antibody | Anti-Syntaxin 6 (rabbit monoclonal) | CST | #2869 | IB 1:1000 |
| Antibody | Anti-LC3B (rabbit monoclonal) | CST | #2775 | IB 1:2000 |
| Antibody | Anti-HA (rabbit monoclonal) | CST | #3724S | IB 1:5000 IF 1:200 |
| Antibody | Anti-PSD95 (mouse monoclonal) | NeuroMab | 75028 | IB 1:5000 |
| Antibody | Anti-FLAG M2 HRP (mouse monoclonal) | Sigma | A8592 | IB 1:5000 |
| Commercial assay or kit | TMT10plex Isobaric Label Reagent Set | Thermo Scientific | 90113 | |
| Commercial assay or kit | Pierce High pH Reversed-Phase Peptide Fractionation Kit | Thermo Scientific | 84868 | |
| Commercial assay or kit | Pierce protein G magnetic beads | Thermo Scientific | 88848 | |
| Commercial assay or kit | Pierce anti-HA magnetic beads | Thermo Scientific | 88837 | |
| Sequence-based reagent | KO sgRNA1 (5'- CGATGCTCCAGGCCACAAGGAGG -3') | *Michlits et al., 2020* | | Targeting exon 5 |
| Sequence-based reagent | KO sgRNA2 (5'-GCCGTGGCAGCATTGGTGTGTGG -3') | *Michlits et al., 2020* | | Targeting exon 3 |
| Sequence-based reagent | KO sgRNA3 (5'- TGGGCCTGGGCAACATCTGGAGG -3') | *Michlits et al., 2020* | | Targeting exon 2 |
| Chemical compound, drug | 3,3-Diaminobenzidine | Sigma | D5905 | |
| Software, algorithm | Noldus Video Tracking Software | Noldus | Ethovision XT 15 | |
| Software, algorithm | Prism 8 | GraphPad Software | | |
| Other | SeQuant ZIC-HILIC column (150 mm × 2.1 mm, 3.5 µm) | Merck Millipore | 150442 | HPLC column for chemical detection |
| Other | SeQuant ZIC-pHILIC column (150 mm × 2.1 mm, 5 µm) | Merck Millipore | 150460 | HPLC column for chemical detection |

