## [Editor Report · eLife assessment]

This study makes a **valuable** contribution to our functional understanding of the atypical amino acid transporter SLC6A177 at nerve cell synapses and the role of SLC6A17 variants in certain forms of intellectual disability. The reported evidence that disease-linked SLC6A17 variants cause behavioral abnormalities is **convincing**. However, corresponding molecular underpinnings, that is, the molecular role of SLC6A17 in synapses and the functional molecular consequences of disease-related SLC6A17 variations, remain unclear because corresponding informative experimental approaches are missing – most importantly direct measurements of the transport activity of SLC6A17 in the various genetic contexts studied. This limits the robustness and validity of key mechanistic conclusions drawn from the present work.

---

## [Referee Report · Joint Public Review]

The molecular composition of synaptic vesicles (SVs) has been defined in substantial detail, but the function of many SV-resident proteins are still unknown. The present study focused on one such protein, the 'orphan' SV-resident transporter SLC6A17. By utilizing sophisticated and extensive mouse genetics and behavioral experiments, the authors provide convincing support for the notion that certain SLC6A17 variants cause intellectual disability (ID) in humans carrying such genetic variations. This is an important and novel finding. Furthermore, the authors propose, based on LC-MS analyses of isolated SVs, that SLC6A17 is responsible for glutamine (Gln) transport into SVs, leading to the provocative idea that Gln functions as a neurotransmitter and that deficits in Gln transport into SVs by SLC6A17 represents a key pathogenetic mechanism in human ID patients carrying variants of the SLC6A17 gene.

This latter aspect of the present paper is not adequately supported by the experimental evidence so that the main conceptual claims of the study appear insufficiently justified at this juncture. Key weaknesses are as follows:

A. Detection of Gln, along with classical neurotransmitters such as glutamate, GABA, or ACh, in isolated SV fractions does not prove that Gln is transported into SVs by active transport. Gln is quite abundant in extracellular compartments. Its appearance in SV samples can therefore also be explained by trapping in SVs during endocytosis, presence in other - contaminating - organelles, binding to membrane surfaces, and other processes. Direct assays of Gln uptake into SVs, which have the potential to stringently test key postulates of the authors, are lacking.

B. The authors generated multiple potentially very useful genetic tools and models. However, the validation of these models is incomplete. Most importantly, it remains unclear whether the different mutations affect SLC6A17 expression levels, subcellular localization, or the expression and trafficking of other SV and synapse components.

C. Apart from the caveats mentioned above regarding Gln uptake into SVs, the data interpretation provided by the authors lacks stringency with respect to the biophysics of plasma membrane and SV transporters.

---

## [Author Response]

**Joint Public Review**
The molecular composition of synaptic vesicles (SVs) has been defined in substantial detail, but the function of many SV-resident proteins are still unknown. The present study focused on one such protein, the 'orphan' SV-resident transporter SLC6A17. By utilizing sophisticated and extensive mouse genetics and behavioral experiments, the authors provide convincing support for the notion that certain SLC6A17 variants cause intellectual disability (ID) in humans carrying such genetic variations. This is an important and novel finding. Furthermore, the authors propose, based on LCMS analyses of isolated SVs, that SLC6A17 is responsible for glutamine (Gln) transport into SVs, leading to the provocative idea that Gln functions as a neurotransmitter and that deficits in Gln transport into SVs by SLC6A17 represents a key pathogenetic mechanism in human ID patients carrying variants of the SLC6A17 gene.This latter aspect of the present paper is not adequately supported by the experimental evidence so that the main conceptual claims of the study appear insufficiently justified at this juncture. Key weaknesses are as follows:A) Detection of Gln, along with classical neurotransmitters such as glutamate, GABA, or ACh, in isolated SV fractions does not prove that Gln is transported into SVs by active transport. Gln is quite abundant in extracellular compartments. Its appearance in SV samples can therefore also be explained by trapping in SVs during endocytosis, presence in other - contaminating - organelles, binding to membrane surfaces, and other processes. Direct assays of Gln uptake into SVs, which have the potential to stringently test key postulates of the authors, are lacking.

We have conducted multiple control experiments to exclude the possibility of contamination.

1). Western blot analysis of SLC6A17-HA immunoisolation (Figure 4D and Figure 4—figure supplement 1) has shown that this faction contained little other organelles and membranes. These results are strong argument that contaminations in our isolated fraction were in very low level.

2). We then examined the proportion of SLC6A17 localized SVs through quantifying the co-localization of Syp and SLC6A17 by anti-Syp immunoisolation in Slc6a17-2A-HA-iCre mice. We found that SLC6A17 is predominately localized on SVs (with 98.7% compared with classical SV marker, Author response image 1A). This further showed that immunoisolated SLC6A17 fraction was mainly composed of SVs.

3). We also analyzed other SV marker proteins such as Syt1 and Syb2 for IP-LC-MS, all results supportedGln enrichment (Author response image 1B).

4). Importantly, immunoisolation of the SLC6A17P633R-HA protein, which caused SLC6A17 mislocalization away from the SVs (Figure 3B and Figure 3—figure supplement 1C, D), showed no Gln enrichment (Author response image 1C).

5). Moreover, immunoisolation of AAV-PHP.eb overexpressed cytoplasmic membrane Gln transporter SLC38A1-HA did not show Gln enrichment (Author response image 1D).

6). We also tested whether trafficking organelles such as the lysosome could enrich Gln. As is shown in Author response image 1E, immunoisolation of AAV-PHP.eb overexpressed TMEM192-HA did not show Gln enrichment.For active transport, we tested the effects of proton dissipator FCCP, v-ATPase inhibitor NEM and ΔpH dissipator nigercin. As is shown in Author response image 1F, 1G, Gln level was reduced by these inhibitors, supporting active transport of Gln.

**Author response image 1. sa2fig1:** Control experiments to test for contamination. A. Anti-Syp immunoisolation in Slc6a17-2A-HA-iCre mice. B. Quantification of Gln level in anti-Syt1 and anti-Syb2 immunoisolated fraction. C. Anti-HA immunoisolation in SLC6A7-2A-HA and anti-Slc6a17P633R mice. D. Anti-HA immunoisolation in AAV-PHP.eb-hSyn-SLC38A1-HA overexperssion mice. E. Anti-HA immunoisolation in AAV-PHP.eb-hSyn-TMEM192-HA overexperssion mice. F. Anti-HA immunoisolation in SLC6A7-2A-HA mice under FCCP (50 μM) and NEM (200 μM). G. Anti-Syp immunoisolation in wild type mice under FCCP (50 μM) and Nigercin (20 μM).

B) The authors generated multiple potentially very useful genetic tools and models. However, the validation of these models is incomplete. Most importantly, it remains unclear whether the different mutations affect SLC6A17 expression levels, subcellular localization, or the expression and trafficking of other SV and synapse components.

The verification of transgenic mouse line is described in the Material and Methods section of our manuscript. There are numerous literatures published for CRISPR mediated gene editing in animals and the off-target effect of CRISPR-Cas9 system is widely studied with optimized design tools developed by many groups ([83]; Chu et al., 2015, 2016; Liu et al., 2017; Gemberling et al., 2021; Singh et al., 2022). The gRNAs used for animal generation were chosen carefully based on publically available tools. Apart from basic genomic PCR sequencing of target regions of all gene edited mouse models, Southern blots were performed by Biocytogen company for Slc6a17-HA-2A-iCre and Slc6a17P633R mice to rule out random insertions. Expression levels in Slc6a17-KO and Slc6a17P633R mice were not affected, as shown in Figure R2. HA-tagged protein in Slc6a17-HA-2A-iCre and Slc6a17P633R mice were detected by immunoisolation, immunofluorescence, and fractionation (Figure 3, 4, Figure 3—figure supplement 1, Figure 4—figure supplement 1). Both showed localizations expected from previous reports ().

C) Apart from the caveats mentioned above regarding Gln uptake into SVs, the data interpretation provided by the authors lacks stringency with respect to the biophysics of plasma membrane and SV transporters.

The biophysics of SLC6A17 was carefully studied (Para et al 2008; Zaia and Reimer, 2009). Our work focused on in vivo biochemical results, not biophysics.

**Author response image 2. sa2fig2:** Verification of genetic mouse models. A. q-PCR verification of Slc6a17-KO mice; B. q-PCR verification of Slc6a17P633R mice; C. Example of genomic primer design for Slc6a17-HA-2A-iCre mice founder mice screen; D. Example of genomic PCR for Slc6a17-HA-2A-iCre mice founder mice screen; E. Southern blot performed for Slc6a17-HA-2A-iCre mice.

Reference

Chu, Van Trung et al. “Increasing the efficiency of homology-directed repair for CRISPR-Cas9-induced precise gene editing in mammalian cells.” Nature biotechnology vol. 33,5 (2015): 543-8. doi:10.1038/nbt.3198

Chu, Van Trung, et al. "Efficient generation of Rosa26 knock-in mice using CRISPR/Cas9 in C57BL/6 zygotes." BMC biotechnology 16.1 (2016): 1-15.

Gemberling, Matthew P et al. “Transgenic mice for in vivo epigenome editing with CRISPR-based systems.” Nature methods vol. 18,8 (2021): 965-974. doi:10.1038/s41592-021-01207-2

Liu, Edison T., et al. "Of mice and CRISPR: The post‐CRISPR future of the mouse as a model system for the human condition." EMBO reports 18.2 (2017): 187-193.

Madisen, Linda, et al. "A robust and high-throughput Cre reporting and characterization system for the whole mouse brain." Nature neuroscience 13.1 (2010): 133-140.

Parra, Leonardo A., et al. "The orphan transporter Rxt1/NTT4 (SLC6A17) functions as a synaptic vesicle amino acid transporter selective for proline, glycine, leucine, and alanine." Molecular pharmacology 74.6 (2008): 15211532.

Platt, R.J., Chen, S., Zhou, Y., Yim, M.J., Swiech, L., Kempton, H.R., Dahlman, J.E., Parnas, O., Eisenhaure, T.M., Jovanovic, M., et al. (2014). CRISPR-Cas9 knockin mice for genome editing and cancer mode Yang, Hui, Haoyi Wang, and Rudolf Jaenisch. "Generating genetically modified mice using CRISPR/Cas-mediated genome engineering." Nature protocols 9.8 (2014): 1956-1968.ling. Cell 159, 440-455.

Singh, Surender et al. “Opportunities and challenges with CRISPR-Cas mediated homologous recombination based precise editing in plants and animals.” Plant molecular biology, 10.1007/s11103-022-01321-5. 31 Oct. 2022, doi:10.1007/s11103-022-01321-5

Zaia, K.A., and Reimer, R.J. (2009). Synaptic vesicle protein NTT4/XT1 (SLC6A17) catalyzes Na+-coupled neutral amino acid transport. J Biol Chem 284, 8439-8448.